# Dual Self-Awareness Value Decomposition Framework without Individual Global Max for Cooperative MARL

**Zhiwei Xu, Bin Zhang, Dapeng Li, Guangchong Zhou, Zeren Zhang, Guoliang Fan**
Institute of Automation, Chinese Academy of Sciences
School of Artificial Intelligence, University of Chinese Academy of Sciences
`{xuzhiwei2019, guoliang.fan}@ia.ac.cn`

## Abstract

Value decomposition methods have gained popularity in the field of cooperative multi-agent reinforcement learning. However, almost all existing methods follow the principle of Individual Global Max (IGM) or its variants, which limits their problem-solving capabilities. To address this, we propose a dual self-awareness value decomposition framework, inspired by the notion of dual self-awareness in psychology, that entirely rejects the IGM premise. Each agent consists of an ego policy for action selection and an alter ego value function to solve the credit assignment problem. The value function factorization can ignore the IGM assumption by utilizing an explicit search procedure. On the basis of the above, we also suggest a novel anti-ego exploration mechanism to avoid the algorithm becoming stuck in a local optimum. As the first fully IGM-free value decomposition method, our proposed framework achieves desirable performance in various cooperative tasks.

## 1 Introduction

Multi-agent reinforcement learning (MARL) has recently drawn increasing attention. MARL algorithms have been successfully applied in diverse fields, including game AI [1] and energy networks [37], to solve practical problems. The fully cooperative multi-agent task is the most common scenario, where all agents must cooperate to achieve the same goal. Several cooperative MARL algorithms have been proposed, with impressive results on complex tasks. Some algorithms [7, 32, 17] enable agents to collaborate through communication. However, these approaches incur additional bandwidth overhead. Most communication-free MARL algorithms follow the conventional paradigm of centralized training with decentralized execution (CTDE) [20]. These works can be roughly categorized into multi-agent Actor-Critic algorithm [8, 16] and value decomposition method [34, 27, 19, 40, 39]. Due to the high sample efficiency in off-policy settings, value decomposition methods can outperform other MARL algorithms.

However, due to the nature of the CTDE paradigm, the value decomposition method can only operate under the assumption of Individual Global Max (IGM) [30]. The IGM principle links the optimal joint action to the optimal individual one, allowing the agent to find the action corresponding to the maximal Q value during decentralized execution. Nevertheless, this assumption is not applicable in most real-world scenarios. And most value decomposition approaches [27] directly constrain the weights of the neural network to be non-negative, which restricts the set of global state-action value functions that can be represented. We confirm this point through experiments in Appendix D.1.

Removing the IGM assumption requires agents to be able to find actions corresponding to the maximal joint Q value during execution. Therefore, in addition to the evaluation network, agents also need an additional network to learn how to *search* for the best actions in a decentralized manner. Similarly, there are concepts of *ego* and *alter ego* in psychology. The ego usually refers to the conscious part of the individual, and Freud considered the ego to be the executive of the personality [9]. Some people

believe that an alter ego pertains to a different version of oneself from the authentic self [24, 12]. Others define the alter ego in more detail as the evaluation of the self by others [3]. As a result, both ego and alter ego influence a person's personality and behaviour, as depicted in Figure 1. Besides, Duval and Wicklund [5] put forth the dual self-awareness hypothesis, which posits that individuals possess two states of consciousness: public self-awareness and private self-awareness. Private self-consciousness refers to an individual's internal understanding of himself, whereas public self-awareness enables individuals to recognize that

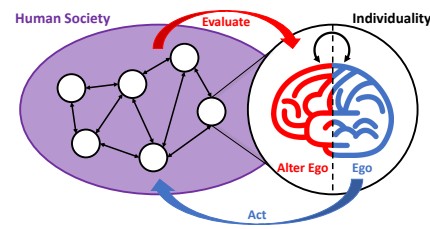

Figure 1: In psychology, it is commonly believed that the ego and alter ego determine an individual's behavior jointly.

they are evaluated by others. Enlightened by these psychological concepts, we propose a novel MARL algorithm, **D**ual self-**A**wareness **V**alue d**E**composition (**DAVE**). Each agent in DAVE maintains two neural network models: the alter ego model and the ego model. The former participates in the evaluation of the group to solve the global credit assignment problem, and the latter is not directly optimized by the joint state-action value. The ego policy can help the alter ego value function to find the individual action corresponding to the optimal joint action through explicit search procedures so that the value decomposition framework can completely eliminate the constraints of the IGM assumption. Furthermore, to prevent the ego policy based on the above framework from potentially getting stuck in a bad local optimum, we also propose an anti-ego exploration method. This method can effectively avoid the underestimation of the optimal action that may be caused by the limited search procedure.

We first evaluate the performance of DAVE in several didactic problems to demonstrate that DAVE can resolve the issues caused by IGM in conventional value decomposition methods. Then we run DAVE in various complex scenarios including StarCraft II, and compare it to other popular baselines. Our experimental results show that DAVE can still achieve competitive performance despite the absence of IGM. To the best of our knowledge, DAVE is the first multi-agent value decomposition method that completely abandons the IGM assumption.

## 2 Related Work

Recently, value decomposition methods have made significant progress in problems involving the decentralized partially observable Markov decision process (Dec-POMDP) [23]. By calculating a joint state-action value function based on the utility functions of all agents, value decomposition methods can effectively alleviate the notorious credit assignment issue. The first proposed method, VDN [34], simply sums up the utility of all agents to evaluate joint actions. This undoubtedly limits the range of joint value functions that can be approximated. QMIX [27] incorporates hypernetworks [11] into the value decomposition method and employs the mixing network to fit the joint value function based on the individual ones, thereby expressing richer classes of joint value function. To enable the agent to choose actions with the greedy policy during decentralized training, QMIX assumes monotonicity and restricts the weight parameters of the mixing network output by hypernetworks to non-negative. So the monotonic mixing network still cannot represent some class of value functions. Various value decomposition variants have subsequently been proposed. Officially proposing IGM, QTRAN [30] indicated that the value decomposition method must satisfy the assumption that the optimal joint actions across agents are equivalent to the collection of individual optimal actions of each agent. Unlike VDN and QMIX, QTRAN is not restricted by structural constraints such as non-negative weights, but it performs poorly in complex environments such as StarCraft II. MAVEN [21] enhances its performance by improving its exploration capabilities. Weighted QMIX [28] takes into account a weighted projection that places greater emphasis on better joint actions. QPLEX [36] takes a duplex dueling network architecture to factorize the joint action-value function and proposes the advantage-based IGM. Nevertheless, the aforementioned methods only loosen the restrictions by proposing IGM variants. More importantly, existing policy-based MARL methods [41, 20, 25] do not explicitly introduce IGM, but they also cannot solve some non-monotonic problems. So how to remove the IGM assumption is still an open and challenging problem. Our proposed DAVE is completely IGM-free and can be applied to existing IGM-based value decomposition methods. More related work will be discussed in Appendix E.

## 3 Preliminaries

### 3.1 Dec-POMDPs

In this paper, we concentrate on the Dec-POMDP problems. It is an important mathematical model for multi-agent collaborative sequential decision-making problem, in which agents act based on different pieces of information about the environment and maximize the global shared return. Dec-POMDPs can be formulated as a tuple $\langle \mathcal{A}, \mathcal{S}, \mathcal{U}, \mathcal{Z}, P, O, r, \gamma \rangle$. $a \in \mathcal{A} := \{1, \ldots, n\}$ represents the agent in the task. $s \in \mathcal{S}$ is the global state of the environment. The joint action set $\mathcal{U}$ is the Cartesian product of action sets $\mathcal{U}^a$ of all agents and $\boldsymbol{u} \in \mathcal{U}$ denotes the joint action. Given the state $s$ and the joint action $\boldsymbol{u}$, the state transition function $P : \mathcal{S} \times \mathcal{U} \times \mathcal{S} \to [0, 1]$ returns the probability of the next potential states. All agents receive their observations $\boldsymbol{z} = \{z^1, ..., z^n\}$ obtained by the observation function $O : \mathcal{S} \times \mathcal{A} \to \mathcal{Z}$. $r : \mathcal{S} \times \mathcal{U} \to \mathbb{R}$ denotes the global shared reward function and $\gamma$ is the discount factor. Besides, recent work frequently allows agents to choose actions based on historical trajectory $\tau^a$ rather than local observation $z^a$ in order to alleviate the partially observable problem.

In Dec-POMDP, all agents attempt to maximize the discount return $R_t = \sum_{t=0}^{\infty} \gamma^t r_t$ and the joint state-action value function is defined as:

$$Q_{\text{tot}}(s, \boldsymbol{u}) = \mathbb{E}_{\boldsymbol{\pi}} \left[ R_t | s, \boldsymbol{u} \right],$$

where $\boldsymbol{\pi} = \{\pi_1, \ldots, \pi_n\}$ is the joint policy of all agents.

### 3.2 IGM-based Value Decomposition Methods

The goal of value decomposition method is to enable the application of value-based reinforcement learning methods in the CTDE paradigm. This approach decomposes the joint state-action value function $Q_{\text{tot}}$ into the utility function $Q_a$ of each agent. In VDN, the joint state-action value function is equal to the sum of all utility functions. And then QMIX maps utility functions to the joint value function using a monotonous mixing network and achieve better performance.

Nonetheless, additivity and monotonicity still impede the expressive capacity of the neural network model, which may result in the algorithm being unable to handle even for some simple matrix game problems. To this end, QTRAN proposes the Individual Global Max (IGM) principle, which can be represented as:

$$\arg\max_{u^a} Q_{\text{tot}}(s, \boldsymbol{u}) = \arg\max_{u^a} Q_a \left( \tau^a, u^a \right), \quad \forall a \in \mathcal{A}.$$

The IGM principle requires consistency between local optimal actions and global ones. Most value decomposition methods including VDN and QMIX must follow IGM and its variants. This is because only when the IGM principle holds, the individual action corresponding to the maximum individual action value during decentralized execution is a component of the optimal joint action. Otherwise, the agent cannot select an appropriate action based solely on the individual action-value function.

## 4 Methodology

In this section, we introduce the implementation of the dual self-awareness value decomposition (DAVE) framework in detail. DAVE incorporates an additional ego policy through supervised learning in comparison to the conventional value decomposition method, allowing the agent to identify the optimal individual action without relying on the IGM assumption. Moreover, DAVE can be applied to most IGM-based value decomposition methods and turn them into IGM-free ones.

### 4.1 Dual Self-Awareness Framework

Almost all value decomposition methods follow the IGM assumption, which means that the optimal joint policy is consistent with optimal individual policies. According to IGM, the agent only needs to follow the greedy policy during decentralized execution to cater to the optimal joint policy. Once the IGM assumption is violated, agents cannot select actions that maximize the joint state-action value based solely on local observations. Consequently, the objective of the IGM-free value decomposition method is as follows:

$$\arg\max_{\boldsymbol{\pi}} Q_{\text{tot}}(s, \boldsymbol{u}), \tag{1}$$

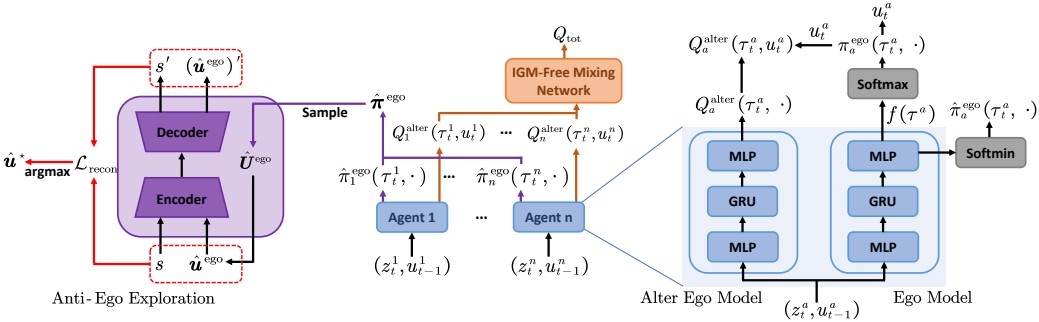

Figure 2: **Middle**: The overall dual self-awareness value decomposition framework. **Right**: Agent network structure. **Left**: The setup of the anti-ego exploration.

where $\boldsymbol{u} = \{u^1, \ldots, u^n\}$ and $u^a \sim \pi_a(\tau^a)$. Without the IGM assumption, Equation (1) is NP-hard because it cannot be solved and verified in polynomial time [14]. Therefore, in our proposed dual self-awareness framework, each agent has an additional policy network to assist the value function network to find the action corresponding to the optimal joint policy. Inspired by the dual self-awareness theory in psychology, we named these two networks contained in each agent as *ego policy model* and *alter ego value function model*, respectively. Figure 2 depicts the dual self-awareness framework. Note that the IGM-free mixing network refers to the mixing network obtained by removing structural constraints. For instance, the IGM-free QMIX variant does not impose a non-negative constraint on the parameters of the mixing network.

In DAVE, the alter ego value function model of each agent $a$ is identical to the agent network in other value decomposition methods and generates individual state-action value $Q_a^{\mathrm{alter}}(\tau^a, \cdot)$ based on the current historical trajectory $\tau^a$. To obtain the joint state-action value $Q_{\mathrm{tot}}^{\mathrm{alter}}(s, \boldsymbol{u})$ for carrying out the joint action $\boldsymbol{u}$ in the state $s$, the value function of each agent is fed into the IGM-free mixing network. However, the alter ego model does not determine the actions chosen by the agent, either during centralized training or decentralized execution. What determines the action of each agent is the ego policy $\pi_a^{\mathrm{ego}}$. The joint ego policy can be expressed as $\boldsymbol{\pi}^{\mathrm{ego}}(\boldsymbol{u} \mid s) = \prod_{a=1}^n \pi_a^{\mathrm{ego}}(u^a \mid \tau^a)$. When executing, the agent interacts with the environment by sampling actions from the categorical distribution output by the ego policy. And during training, we replace the exact maximization of $Q_{\mathrm{tot}}(s, \boldsymbol{u})$ over the total joint action space with maximization over a set of samples $\boldsymbol{U}^{\mathrm{ego}}$ from $\boldsymbol{\pi}^{\mathrm{ego}}$. So we modify the objective of the cooperative multi-agent value decomposition problem from Equation (1) to

$$\arg\max_{\boldsymbol{\pi}^{\mathrm{ego}}} Q_{\mathrm{tot}}^{\mathrm{alter}}(s, \boldsymbol{u}^{\mathrm{ego}}), \qquad \text{s.t.} \quad \boldsymbol{u}^{\mathrm{ego}} \in \boldsymbol{U}^{\mathrm{ego}}, \tag{2}$$

where $\boldsymbol{U}^{\mathrm{ego}} := \{\boldsymbol{u}_i^{\mathrm{ego}} \sim \boldsymbol{\pi}^{\mathrm{ego}}(s)\}_{i=1}^M$ in state $s$ and $M$ is the number of samples. Therefore, the dual self-awareness framework substitutes the global optimal state-action value with the local optimal one found in the small space $\boldsymbol{U}^{\mathrm{ego}}$, which is obtained by sampling $M$ times from $\boldsymbol{\pi}^{\mathrm{ego}}$. As long as the ego policy assigns non-zero probabilities to all available actions, this method will converge to the objective described by Equation (1) as the number of samples $M$ increases. The detailed proof can be found in Appendix A. Nevertheless, the computational cost is directly proportional to $M$. So the choice of $M$ must strike a balance between precision and computational complexity.

Next, we introduce the update process of the alter ego value function and the ego policy in the dual self-awareness framework. According to the modified objective of the cooperative multi-agent value decomposition problem illustrated by Equation (2), the ego policy needs to be able to predict the individual action that corresponds to the maximal joint state-action value. Therefore, we update the ego policy through supervised learning to increase the probability of the optimal joint action obtained through the sampling procedure. We define $\boldsymbol{u}^\star = \arg\max_{\boldsymbol{u}^{\mathrm{ego}}} Q_{\mathrm{tot}}^{\mathrm{alter}}(s, \boldsymbol{u}^{\mathrm{ego}})$ as the joint action corresponding to the maximal joint state-action value in the samples, where $\boldsymbol{u}^{\mathrm{ego}} \in \boldsymbol{U}^{\mathrm{ego}}$. The loss function for the joint ego policy can be written as:

$$\mathcal{L}_{\mathrm{ego}} = -\log \boldsymbol{\pi}^{\mathrm{ego}}(\boldsymbol{u}^\star \mid s). \tag{3}$$

The objective function is the negative log-likelihood of the probability of the joint action $\boldsymbol{u}^\star$ and can be further expressed as $-\sum_{a=1}^n \log \pi_a^{\mathrm{ego}}(u^{a\star} \mid \tau^a)$ associated with individual ego policies.

Additionally, the alter ego value function $\boldsymbol{Q}^{\text{alter}}$ is constantly changing during the training process. To prevent the ego policy $\boldsymbol{\pi}^{\text{ego}}$ from becoming too deterministic and getting trapped in local optimal solutions, we will introduce *a novel exploration method* in the next subsection.

Conventional value decomposition methods can be trained end-to-end by minimizing the following loss:

$$\mathcal{L} = (y_{\text{tot}} - Q_{\text{tot}}(s_t, \boldsymbol{u}_t))^2, \tag{4}$$

where $y_{\text{tot}} = r_t + \gamma \max_{\boldsymbol{u}_{t+1}} Q_{\text{tot}}^-(s_{t+1}, \boldsymbol{u}_{t+1})$ and $Q_{\text{tot}}^-$ is the target joint value function. The loss function described above cannot be directly applied to our proposed dual self-awareness framework because of the max operator. If we also traverse the samples of the joint action in $\boldsymbol{U}^{\text{ego}}$ to select the local maximum target joint state-action value function, which may result in high variance. Inspired by the Expected SARSA [35], we use the expected value instead of the maximal one. The update rule of $Q_{\text{tot}}^{\text{alter}}$ considers the likelihood of each action under the current policy and can eliminate the variance, which can be expressed as:

$$\mathcal{L}_{\text{alter}} = (y_{\text{tot}}^{\text{alter}} - Q_{\text{tot}}^{\text{alter}}(s_t, \boldsymbol{u}_t))^2, \tag{5}$$

where $y = r_t + \gamma \mathbb{E}_{\boldsymbol{\pi}^{\text{ego}}}[(Q_{\text{tot}}^{\text{alter}})^-(s_{t+1}, \boldsymbol{u}_{t+1})]$. However, this method is still computationally complex. So we reuse the action sample set $\boldsymbol{U}^{\text{ego}}$ and replace $\mathbb{E}_{\boldsymbol{\pi}^{\text{ego}}}[(Q_{\text{tot}}^{\text{alter}})^-(s_{t+1}, \boldsymbol{u}_{t+1})]$ with $\frac{1}{M}\sum_{i=1}^{M}(Q_{\text{tot}}^{\text{alter}})^-(s_{t+1}, \boldsymbol{u}_i^{\text{ego}})$, where $\boldsymbol{u}_i^{\text{ego}} \in \boldsymbol{U}^{\text{ego}}$ in state $s_{t+1}$. The modified update rule for $Q_{\text{tot}}^{\text{alter}}$ reduces the variance and does not increase the order of the computational complexity compared to the original update method shown in Equation (4). Each hypernetwork takes the global state and the actions of all agents as input and generates the weights of the mixing network. It should also be noted that we use the global state and the actions of all agents as input to the hypernetwork that generates weights to enhance the expressiveness of the mixing network.

In summary, we have constructed the fundamental framework of DAVE. Although DAVE consists of a value network and a policy network, it is **DISTINCT** from the Actor-Critic framework or other policy-based methods such as MAPPO [41], MADDPG [20] and FACMAC [25]. DAVE requires an explicit sampling procedure to update the network model. Besides, the actions sampled from the ego policies remain independent and identically distributed (IID) [26], so the ego policy is trained through *supervised learning* rather than *policy gradient* used in policy-based methods. Most intuitively, conventional policy-based methods cannot solve simple non-monotonic matrix games but DAVE can easily find the optimal solution, as described in Section 5.1 and Appendix D.2.

## 4.2 Anti-Ego Exploration

As stated previously, updating the ego policy network solely based on Equation (3) with an insufficient sample size $M$ may lead to a local optimum. This is because the state-action value corresponding to the optimal joint action may be grossly underestimated during initialization, making it challenging for the ego policy to choose the optimal joint action. Consequently, the underestimated $Q_{\text{tot}}^{\text{alter}}$ will have no opportunity to be corrected, creating a vicious circle. Therefore, we developed an *anti-ego exploration* method based on ego policies. Its core idea is to assess the occurrence frequency of state-action pairs using the reconstruction model based on the auto-encoder [13]. The auto-encoder model often requires the encoder to map the input data to a low-dimensional space, and then reconstructs the original input data based on the obtained low-dimensional representation using the decoder. Our method leverages the fact that the auto-encoder cannot effectively encode novel data and accurately reconstruct it. The more significant the difference between the original state-action pair $(s, \boldsymbol{u})$ and the reconstructed output $(s', \boldsymbol{u}')$, the less frequently the agents carry out the joint action $\boldsymbol{u}$ in state $s$. Simultaneously, the difference between the reconstructed state-action pairs and the original ones is also the objective function for updating the auto-encoder, which can be expressed as:

$$\mathcal{L}_{\text{recon}}(s, \boldsymbol{u}) = \text{MSE}(s, s') + \sum_{a=1}^{n} \text{CE}(u^a, (u^a)'), \tag{6}$$

where $\text{MSE}(\cdot)$ and $\text{CE}(\cdot)$ represent the mean square error and the cross-entropy loss function respectively. So we need to encourage agents to choose joint actions with a higher $\mathcal{L}_{\text{recon}}$. Obviously, it is impractical to traverse all available joint actions $\boldsymbol{u}$. Then we look for the most novel actions within the limited set of actions obtained via the sampling procedure. In order to get state-action pairs that are as uncommon as possible, we generate an anti-ego policy $\hat{\boldsymbol{\pi}}^{\text{ego}}$ based on $\boldsymbol{\pi}^{\text{ego}}$, as shown

in Figure 2. The relationship between the ego and anti-ego policy of each agent is as follows:

$$\pi_a^{\text{ego}}(\tau^a) = \text{Softmax}(f(\tau^a)),$$
$$\hat{\pi}_a^{\text{ego}}(\tau^a) = \text{Softmin}(f(\tau^a)), \tag{7}$$

where $f(\cdot)$ denotes the neural network model of ego policy. Equation (7) demonstrates that actions with lower probabilities in $\pi_a^{\text{ego}}$ have higher probabilities in $\hat{\pi}_a^{\text{ego}}$. Of course, actions with low probabilities in $\pi_a^{\text{ego}}$ may be non-optimal actions that have been fully explored, so we filter out such actions through the reconstruction model introduced above. $\hat{\boldsymbol{U}}^{\text{ego}} := \{\hat{\boldsymbol{u}}_i^{\text{ego}} \sim \hat{\boldsymbol{\pi}}^{\text{ego}}(s)\}_{i=1}^{M}$ is the joint action set obtained by sampling $M$ times from the anti-ego policy in state $s$. We can identify the most novel joint action $\hat{\boldsymbol{u}}^\star$ in state $s$ by the following equation:

$$\hat{\boldsymbol{u}}^\star = \arg\max_{\hat{\boldsymbol{u}}^{\text{ego}}} \mathcal{L}_{\text{recon}}(s, \hat{\boldsymbol{u}}^{\text{ego}}), \tag{8}$$

where $\hat{\boldsymbol{u}}^{\text{ego}} \in \hat{\boldsymbol{U}}^{\text{ego}}$. Therefore, the objective function in Equation (3) needs to be modified to increase the probability of the joint action $\hat{\boldsymbol{u}}^\star$. The final objective function of the ego policy $\boldsymbol{\pi}^{\text{ego}}$ can be written as:

$$\mathcal{L}_{\text{ego}} = -\big(\log \boldsymbol{\pi}^{\text{ego}}(\boldsymbol{u}^\star \mid s) + \lambda \log \boldsymbol{\pi}^{\text{ego}}(\hat{\boldsymbol{u}}^\star \mid s)\big), \tag{9}$$

where the exploration coefficient $\lambda \in [0, 1]$ is annealed linearly during the exploration phase. The choice of hyperparameter $\lambda$ requires a compromise between exploration and exploitation. Note that the dual self-awareness framework and the anti ego exploration method are **NOT** two separate contributions. The latter is customized for the former, and it is only used to solve the convergence difficulties caused by the explicit sampling procedure in DAVE. By incorporating the anti-ego exploration module, the dual self-awareness framework with low $M$ can also fully explore the action space and significantly avoid becoming stuck in bad local optima. The whole workflow of the anti-ego exploration mechanism is shown in Appendix B.

Our proposed DAVE can be applied to most existing value decomposition methods, as it simply adds an extra ego policy network to the conventional value decomposition framework. We believe that the IGM-free DAVE framework can utilize richer function classes to approximate joint state-action value functions and thus be able to solve the non-monotonic multi-agent cooperation problems.

## 5 Experiment

In this section, we apply the DAVE framework to standard value decomposition methods and evaluate its performance on both didactic problems and complex Dec-POMDPs. First, we will compare the performance of the DAVE variant with QPLEX, Weighted QMIX, QTRAN, FACMAC-nonmonotonic [25], MAVEN, and the vanilla QMIX. Note that most baselines are directly related to IGM. Then the contribution of different modules in the framework and the impact of hyperparameters will be investigated. Except for the new hyperparameters introduced in the DAVE variant, all other hyperparameters are consistent with the original method. Unless otherwise stated, DAVE refers to the DAVE variant of QMIX, which enhances representational capacity while ensuring fast convergence. For the baseline methods, we obtain their implementation from the released code. The details of the implementation and all environments are provided in Appendix C and Appendix D, respectively.

### 5.1 Single-State Matrix Game

Certain types of tasks in multi-agent cooperation problems cannot be handled under the IGM assumptions, such as the two matrix games we will introduce next. These two matrix games capture very simple cooperative multi-agent tasks and are shown in Figure 3. These payoff matrices of the two-player three-action matrix games describe scenarios that IGM-based value decomposition methods cannot address effectively. The variable $k$ is related to difficulty. In

| $u^1 \backslash u^2$ | A | B | C |
|---|---|---|---|
| A | 8 | -12 | -12 |
| B | -12 | $k$ | 0 |
| C | -12 | 0 | $k$ |

(a) Matrix game I.

| $u^1 \backslash u^2$ | A | B | C |
|---|---|---|---|
| A | $-k$ | 0 | 10 |
| B | 0 | 2 | 0 |
| C | 8 | 0 | $-k$ |

(b) Matrix game II.

Figure 3: Payoffs of the two matrix games.

matrix game I with $k \in [0, 8)$, agents need to choose the same action to get the reward and the optimal joint action is $(A, A)$. Furthermore, the closer $k$ is to 8, the more challenging it becomes to jump out of the local optimum. Similarly, $k \geq 0$ and $(A, C)$ is the optimal joint action in matrix

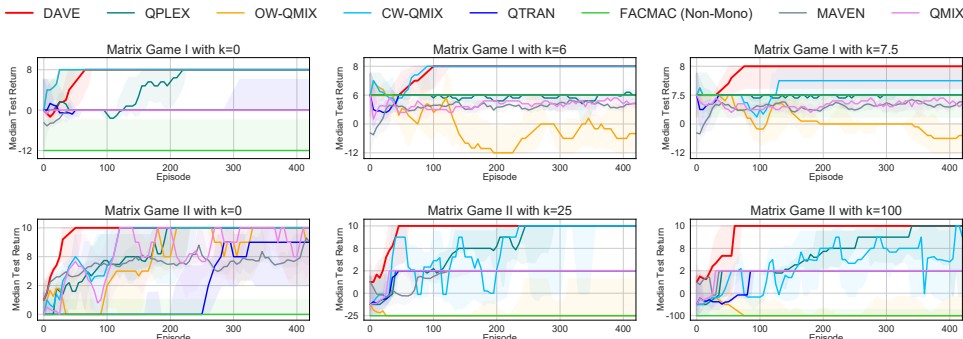

Figure 4: The learning curves of DAVE and other baselines on the matrix games. Note that the ordinates are non-uniform.

game II. Nevertheless, as $k$ increases, so does the risk of selecting the optimal action $(A, C)$, which leads agents in the IGM-based method to prefer the suboptimal joint action $(B, B)$ corresponding to the maximum expected return. Therefore, both matrix games are challenging for traditional value factorization methods. To disregard the influence of exploration capabilities, we implement various algorithms under uniform visitation, which means that all state-action pairs are explored equally. We also evaluate the performance of popular policy-based MARL algorithms. Results in Appendix D.2 demonstrate that although they do not explicitly introduce IGM, policy-based methods still perform poorly on these two simple tasks.

The results of all algorithms on the two matrix games with varying $k$ are presented in Figure 4. we limit the number of interactions between agents and the environment to 400 across all environments. All baselines find it challenging to discover the optimal joint action with such limited data. Compared to other algorithms, DAVE can quickly converge to the optimal solution in all situations and exhibits robust performance, showing little variation as task difficulty increases. CW-QMIX can converge to the optimal joint solution in all cases except matrix game I with $k = 7.5$. QPLEX performs better in matrix game II than in matrix game I, likely due to its sensitivity to the penalty of non-cooperation. In addition, as the task difficulty increases, both CW-QMIX and QPLEX show a significant drop in performance. Although QTRAN can converge to the optimal solution in matrix game I with $k = 0$, it may require more interactive samples for further learning. The remaining algorithms can converge to the optimal joint action $(A, C)$ or the suboptimal joint action $(C, A)$ in matrix game II with $k = 0$, but in the more difficult matrix game II can only converge to the joint action $(B, B)$. We add additional action information to the mixing network of QMIX, similar to that in DAVE, but QMIX still cannot converge to the optimal solution. In summary, DAVE can easily solve the above single-state matrix games and outperforms other IGM-based variants and the basic algorithm QMIX by a significant margin.

## 5.2 Multi-Step Matrix Game

Next, we evaluate the performance of each algorithm in the multi-step matrix game. The environment involves two agents, each with two actions. The initial state is non-monotonic, and agents terminate immediately after receiving a zero reward. If the game is not terminated after the first step, agents need to continue to repeat the last joint action to obtain a non-zero reward. According to Figure 5, agents repeat either the upper-left joint action or the lower-right joint action eight times, resulting in two distinct terminal states. The optimal policy is

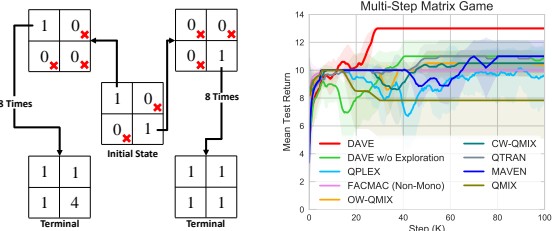

Figure 5: **Left**: Illustrations of the multi-step matrix game. **Right**: Performance over time on the multi-step matrix game.

for agents to repeat the upper-left joint action in the initial state and choose the lower-right joint action in the terminal state, which yields the highest return of 13. Local observations of agents in this

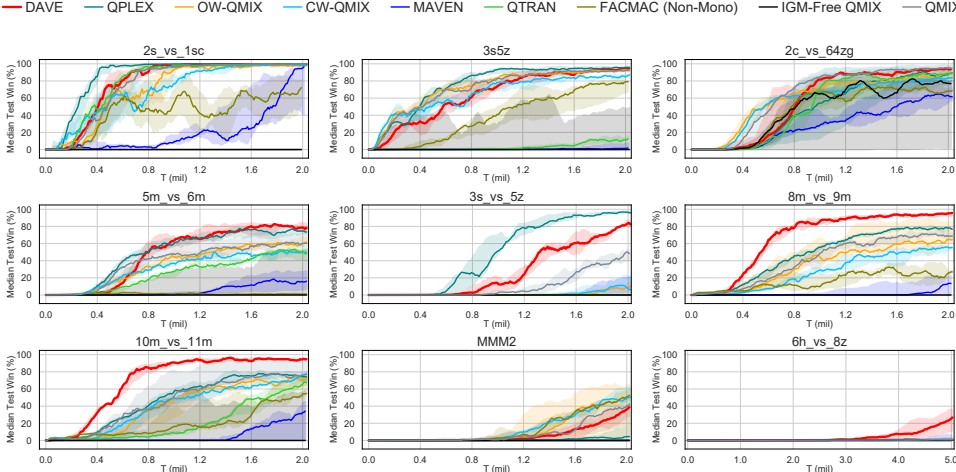

Figure 6: Performance comparison with baselines in different scenarios.

environment include the current step count and the last joint action. The protracted decision-making process requires agents to fully explore the state-action space to find the optimal policy. Therefore, this task can also test the exploration capacity of the algorithm.

Figure 5 shows the performance of DAVE and other baselines on the multi-step matrix game. Only DAVE successfully learned the optimal policy, while most other algorithms were trapped in the suboptimal policy with payoff 10. QPLEX can occasionally converge to the optimal policy, but in most cases it still cannot learn the optimal solution. Besides, to examine the contribution of the anti-ego exploration to the DAVE framework, we implemented DAVE without exploration in this environment. The experimental results show that DAVE without anti-ego exploration still performs better than other baselines, but is significantly inferior to DAVE. Therefore, for some complex tasks, the anti-ego exploration mechanism can assist the agent in exploring novel state-action pairs and avoiding being restricted to suboptimal solutions.

## 5.3 StarCraft II

SMAC [29] is the most popular experimental platform for multi-agent micromanagement, based on the famous real-time strategy game StarCraft II. SMAC offers a diverse range of mini-scenarios, including homogeneous, heterogeneous, symmetric, and asymmetric problems. The goal of each task is to direct agents to defeat enemy units controlled by built-in heuristic AI. Agents can only access information within their own visual range and share a reward function. So we assess the ability of DAVE to solve complex multi-agent cooperation problems in this environment.

The version of StarCraft II used in this paper is 4.10 and performance is not comparable between versions. We test the proposed method in different scenarios. In addition to the above-mentioned methods, baselines also include IGM-Free QMIX that simply does not take the absolute value of the weight of the mixing network and keeps others unchanged. Both IGM-Free QMIX and FACMAC-nonmonotonic are IGM-free methods, but do not have convergence guarantees. The performance comparison between above two IGM-free methods and DAVE that is also IGM-free can reflect the necessity of our proposed ego policy network. In addition, to balance exploration and exploitation, we disable the anti-ego exploration mechanism in some easy scenarios and use it only in hard scenarios.

The experimental results are shown in Figure 6. Undoubtedly, IGM-Free QMIX and FACMAC-nonmonotonic fail to solve almost all tasks. Without the IGM assumption, QMIX is unable to find the correct update direction. Therefore, the assistance of the ego policy network in the DAVE framework is essential for the convergence of IGM-free algorithms. QPLEX performs best in some easy scenarios but fails to solve the task in most hard scenarios. Weighted QMIX performs well in the *MMM2* scenario but has mediocre performance in most other scenarios. Although DAVE is not the best in easy maps, it still shows competitive performance. In hard scenarios, such as *5m_vs_6m*, *8m_vs_9m* and *10m_vs_11m*, DAVE performs significantly better than other algorithms. We believe

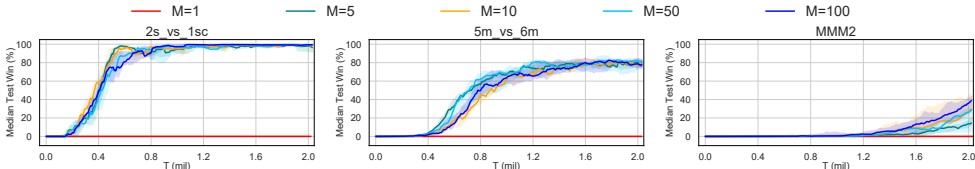

Figure 7: Influence of the sample size $M$ for DAVE.

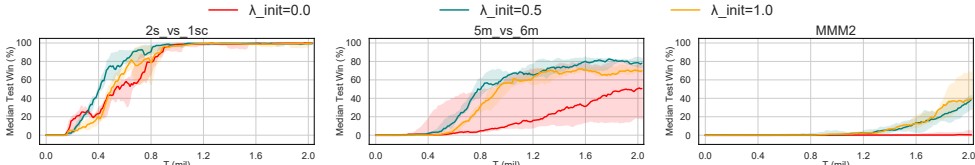

Figure 8: Results of DAVE with different $\lambda_{\text{init}}$. The action space for the three scenarios gradually increases from left to right.

that it may be that the characteristics in these scenarios are far from the IGM assumptions, resulting in low sample efficiency of conventional value decomposition methods. Moreover, because DAVE is IGM-free, it can guide the agent to learn the excellent policy quickly without restriction. Especially in the *6h_vs_8z* scenario, only DAVE got a non-zero median test win ratio. In summary, even if the IGM principle is completely abandoned, DAVE can still successfully solve complex tasks and even outperforms existing IGM-based value decomposition algorithms in certain challenging scenarios. **MORE** experimental results on other complex environments such as SMACv2 [6] and Multi-Agent MuJoCo [25] can be found in Appendix D. These results demonstrate the excellent generalization of DAVE to different algorithms and environments.

### 5.4 Ablation Study

Finally, we evaluate the performance of DAVE under different hyperparameter settings in SMAC scenarios. We focus on the number of samples $M$ and the initial value of the exploration coefficient $\lambda_{\text{init}}$. According to Appendix A, increasing the number of samples $M$ improves the probability of drawing the optimal joint action. Nonetheless, due to time and space complexity constraints, it is crucial to find an appropriate value for the hyperparameter $M$. Then we test DAVE with different $M$ values in three scenarios and present results in Figure 7. The results indicate that a larger $M$ leads to a faster learning speed for DAVE, which aligns with our intuition and proof. DAVE with $M = 1$ fails in all scenarios. When $M$ increases to 10, the performance of DAVE is already considerable. However, as $M$ increases further, the performance enhancement it provides diminishes.

The equilibrium between exploration and exploitation is directly related to the initial value of the exploration coefficient $\lambda_{\text{init}}$. The final value $\lambda_{\text{end}}$ and the annealing period also affect the exploration, but only $\lambda_{\text{init}}$ is discussed in this paper. $\lambda_{\text{end}}$ is generally set to 0, so DAVE is only affected by anti-ego exploration in the early stages of training. Figure 8 shows the learning curve of DAVE under different $\lambda_{\text{init}}$. $\lambda_{\text{init}} = 0$ means that DAVE does not perform anti-ego exploration. The addition of anti-ego exploration in easy scenarios will prevent the algorithm from exploiting better in the early stages, which will significantly slow down the training speed. Conversely, DAVE with high $\lambda_{\text{init}}$ values can achieve superior performance in hard scenarios. So a moderately high $\lambda_{\text{init}}$ is beneficial for DAVE to visit key state-action pairs and converge to the optimal solution. In summary, appropriate values of $M$ and $\lambda_{\text{init}}$ are indispensable for DAVE.

It would be valuable to explore the performance of DAVE with a monotonic mixing network. As shown in Figure 9 and Figure 10, DAVE with a monotonic mixing network still cannot converge to the global optimal solution in matrix games. Since the ego policy is updated by an explicit search procedure, DAVE with a monotonic mixing network has more chances to converge to the optimal solution than vanilla QMIX. However, limited by fewer function families that can be represented, the performance of DAVE with a monotonic mixed network is much worse than that of DAVE and is unstable. In addition, in the complex SMAC environment, the performance of DAVE with a monotonic mixing network is similar to or even better than that of QMIX, but worse than that of

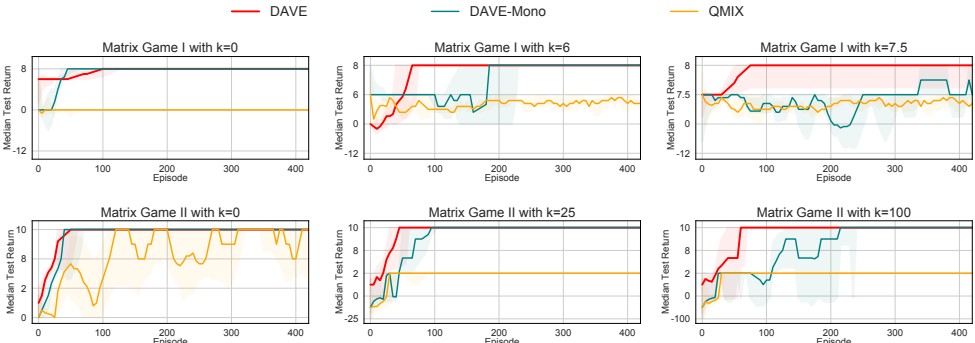

Figure 9: The learning curves of vanilla DAVE and DAVE with a monotonic mixing network on the matrix games.

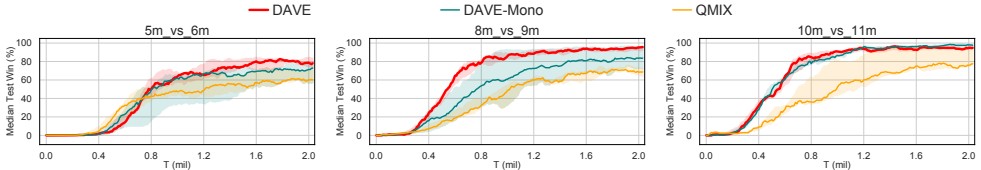

Figure 10: The learning curves of vanilla DAVE and DAVE with a monotonic mixing network on SMAC.

vanilla DAVE. This is also substantial proof that the unconstrained factored mixing network indeed enhances performance.

## 6   Conclusion and Discussion

Inspired by the concept of dual self-awareness in psychology, we propose the IGM-free value decomposition framework DAVE in this paper. By dividing the agent network into two parts, ego policy and alter ego value function, DAVE is the first value decomposition method that completely abandons the IGM assumption. We find that the larger the number of samples in the explicit search procedure, the better the algorithm converges to the global optimum, but this will lead to high computational complexity. Therefore, we propose the anti-ego exploration mechanism, which can effectively prevent DAVE from falling into local optima, especially in problems with large action spaces. Experimental results in some didactic problems and complex environments show that DAVE outperforms existing IGM-based value decomposition methods. We believe that DAVE provides a new perspective on addressing the problems posed by the IGM assumption in value decomposition methods. In our future research, we intend to concentrate on how to choose the exploration coefficients in an adaptive manner. Further study of the issue would be of interest.

## Acknowledgments and Disclosure of Funding

This project was supported by the Strategic Priority Research Program of the Chinese Academy of Sciences, Grant No. XDA27050100.

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

# A  Proof

**Proposition A.1.** *As long as the ego policy assigns non-zero probabilities to all actions, this method will approach the objective described by Equation* (1) *as the number of samples $M$ increases.*

*Proof.* Let $(u^a)^*$ denote the individual actions corresponding to the global optimal joint state-action value function under state $s$, and the optimal joint action is $\boldsymbol{u}^* = \{(u^1)^*, \ldots, (u^n)^*\}$. Then the probability that the sampling procedure draws $\boldsymbol{u}^*$ is expressed as:

$$p(\boldsymbol{u}^*) = 1 - (1 - \boldsymbol{\pi}^{\mathrm{ego}}(\boldsymbol{u}^*|s))^M$$
$$= 1 - (1 - \prod_{i=1}^{n} \pi_a^{\mathrm{ego}}((u^a)^*|\tau^a))^M,$$

where $\pi_a^{\mathrm{ego}}(\cdot|\cdot) \in (0, 1)$ is true for any action. So the second term $(1 - \boldsymbol{\pi}^{\mathrm{ego}}(\boldsymbol{u}^*|s))^M \in (0, 1)$ in the equation decreases as $M$ increases, indicating that $p(\boldsymbol{u}^*)$ is positively correlated with the sample size $M$. $\qquad\square$

# B  Details of the Ego Policy Update

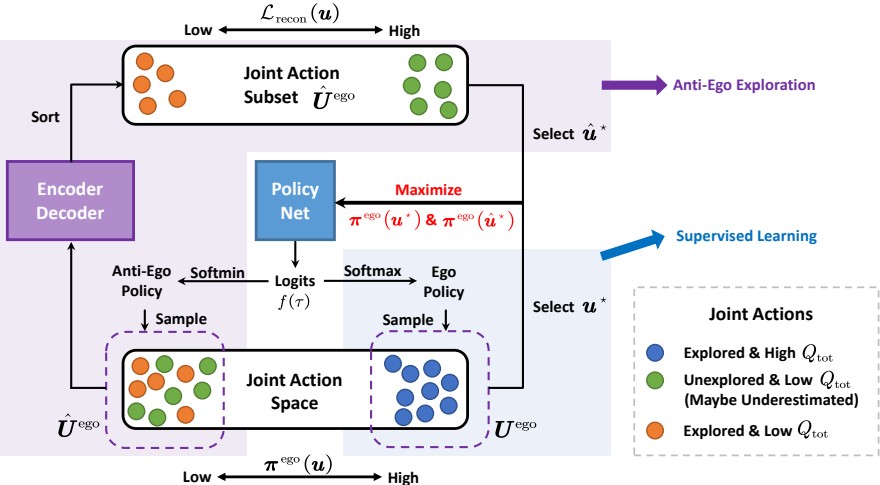

Figure 11: Diagram of the update of ego policy, where $\boldsymbol{U}^{\mathrm{ego}} := \{\boldsymbol{u}_i^{\mathrm{ego}} \sim \boldsymbol{\pi}^{\mathrm{ego}}(s)\}_{i=1}^{M}$ and $\hat{\boldsymbol{U}}^{\mathrm{ego}} := \{\hat{\boldsymbol{u}}_i^{\mathrm{ego}} \sim \hat{\boldsymbol{\pi}}^{\mathrm{ego}}(s)\}_{i=1}^{M}$. Dots indicate different joint actions. The selected actions are $\boldsymbol{u}^\star = \arg\max_{\boldsymbol{u}^{\mathrm{ego}}} Q_{\mathrm{tot}}^{\mathrm{alter}}(s, \boldsymbol{u}^{\mathrm{ego}})$ and $\hat{\boldsymbol{u}}^\star = \arg\max_{\hat{\boldsymbol{u}}^{\mathrm{ego}}} \mathcal{L}_{\mathrm{recon}}(s, \hat{\boldsymbol{u}}^{\mathrm{ego}})$.

The update process of ego policy is shown in Figure 11. During the initialization of neural networks, the state-action value corresponding to the optimal joint action may be underestimated, making it difficult for the ego policy to choose and then correct it, especially in problems with large action spaces. So we use the $\mathrm{Softmin}$ operator to generate an anti-ego policy so that low-probability actions are more likely to be selected. However, not all low-probability joint actions are unexplored and undervalued; some are sub-optimal actions that have been exhaustively explored (orange dots in Figure 11). In order to select only unexplored joint actions, we distinguish the above two through the auto-encoder. Figure 5 and Figure 8 demonstrate the contribution of the anti-ego exploration mechanism.

# C  Implementation Details

## C.1  Algorithmic Description

The pseudo-code of DAVE is shown in Algorithm 1.

**Algorithm 1** Training Procedure for DAVE
___
**Hyperparameters**: Sample size $M$, discount factor $\gamma$, exploration coefficients $\lambda_{\text{init}}$
Initialize the parameters of the neural networks shown in Figure 2
 1: **for** each episode **do**
 2:     Get the global state $s_1$ and the local observations $\boldsymbol{z}_1 = \{z_1^1, z_1^2, \ldots, z_1^n\}$ of all agents
 3:     **for** $t \leftarrow 1$ to $T - 1$ **do**
 4:       **for** $a \leftarrow 1$ to $n$ **do**
 5:         Select action $u_t^a$ according to the ego policy $\pi_a^{\text{ego}}$
 6:       **end for**
 7:       Carry out the joint action $\boldsymbol{u}_t = \{u_t^1, \ldots, u_t^n\}$
 8:       Get the global reward $r_{t+1}$, the next local observations $\boldsymbol{z}_{t+1}$, and the next state $s_{t+1}$
 9:     **end for**
10:     Store the episode in the replay buffer $\mathcal{D}$
11:     Sample a batch of episodes $\mathcal{B} \sim \text{Uniform}(\mathcal{D})$
12:     Sample and obtain the joint action set $\boldsymbol{U}^{\text{ego}} := \{\boldsymbol{u}_i^{\text{ego}} \sim \boldsymbol{\pi}^{\text{ego}}(s)\}_{i=1}^M$ for each trajectory in $\mathcal{B}$
13:     Update the parameters of the alter ego value function and the IGM-free mixing network
        according Equation (5)
14:     Obtain the anti-ego policies $\hat{\pi}_a^{\text{ego}}$ of each agent
15:     Sample and obtain $\hat{\boldsymbol{U}}^{\text{ego}} := \{\hat{\boldsymbol{u}}_i^{\text{ego}} \sim \hat{\boldsymbol{\pi}}^{\text{ego}}(s)\}_{i=1}^M$ by sampling $M$ times from the anti-ego
        policy for each state $s$ in $\mathcal{B}$
16:     Find the most novel joint action $\boldsymbol{u}^\star$ for each state $s$ in $\mathcal{B}$ according Equation (8)
17:     Update the parameters of the ego policy according Equation (9)
18:     Update the parameters of the auto-encoder according Equation (6)
19:     Update the parameters of the target network periodically
20: **end for**
___

## C.2 Hyperparameters

Unless otherwise stated, the hyperparameter settings in different environments are shown in Table 1, which are exactly the same as the settings in PyMARL[1]. All experiments in this paper are run on Nvidia GeForce RTX 3090 graphics cards and Intel(R) Xeon(R) Platinum 8280 CPU. For baselines, exploration is performed during training using independent $\epsilon$-greedy action selection. $\epsilon$ is annealing linearly from 1.0 to 0.05 and the annealing period of $\epsilon$ is 50000 steps. All experiments in this paper are carried out with 6 random seeds in order to avoid the effect of any outliers.

| Name | Description | Value |
|------|-------------|-------|
|  | Learning rate | 0.0005 |
|  | Type of optimizer | RMSProp |
| $\alpha$ | RMSProp param | 0.99 |
| $\epsilon$ for optimizer | RMSProp param | 0.00001 |
|  | How many episodes to update target networks | 200 |
|  | Reduce global norm of gradients | 10 |
|  | Batch size | 32 |
|  | Capacity of replay buffer | 5000 |
| $\gamma$ | Discount factor | 0.99 |
| $M$ | Sample size | 100 |
| $\lambda_{\text{init}}$ | Starting value for exploration coefficient annealing | 0.5 |
| $\lambda_{\text{end}}$ | Ending value for exploration coefficient annealing | 0 |
|  | Annealing period for the multi-step matrix game | 25k |
|  | Annealing period for easy scenarios in SMAC | 500k |
|  | Annealing period for hard scenarios in SMAC | 1.5m |

Table 1: Hyperparameter settings.

___
[1]https://github.com/oxwhirl/pymarl.

## D   Additional Experiments and Details

### D.1   Comparison between IGM-Based and IGM-Free Mixing Networks

To verify that IGM-based mixing network restricts the set of global state-action value functions that can be represented, we synthesize an artificial dataset to represent the reward functions of two agents in a single state matrix game. Each agent has 101 actions, which means $u^i \in \{0, 1, \ldots, 100\}$. In order to only measure the influence of whether the mixing network follows the IGM assumption, we fix the individual action-value function as follows:

$$Q_i(u^i) = \frac{u^i}{100}, \quad \forall i \in \{1, 2\}.$$

The reward value $R$ for a joint action is given as follows:

$$R(u^1, u^2) = \sin\left(2\pi Q_1(u^1)\right) + \exp\left(Q_2(u^2)\right).$$

The ground-truth joint value function is shown in Figure 12. Next we calculate the joint action value $Q_{\text{tot}}$ for the selected $\boldsymbol{u}$ based on the given $Q_i$ and $R$. Most value decomposition approaches, such as QMIX and QPLEX, directly constrain the weights of the neural network to be non-negative, so we use the mixing network in QMIX as the IGM-based model to be trained. Meanwhile, the variant that removes the absolute function is viewed as the IGM-free model. It should also be noted that we use the global state and the actions of all agents as input to the hypernetwork that generates weights to enhance the expressiveness of the mixing network. Figure 12 represents the loss curve and Figure 13 illustrates the joint action-value function estimated by the two models.

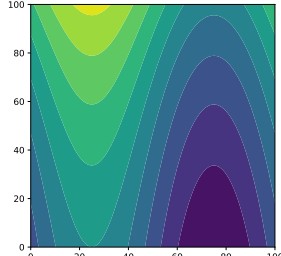
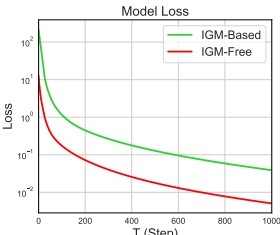

Figure 12: **Left**: Contours for ground-truth $Q_{\text{tot}}$, where $x$-axis and $y$-axis mean each agent's actions. **Right**: The loss curves for two models during training.

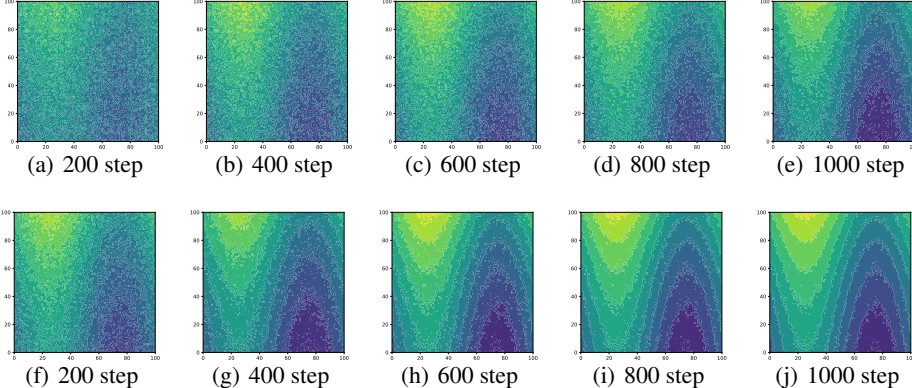

Figure 13: Contours for estimated $Q_{\text{tot}}$. Colored values represent the values of $Q_{\text{tot}}$ for IGM-based ((a)-(e)) and IGM-free ((f)-(j)) mixing networks.

From the results, it can be concluded that when the joint value function $Q_{\text{tot}}$ is not a simple monotone function with respect to $Q_i$, the representation ability of the IGM-based mixing network is obviously limited, and the convergence speed is also slower than IGM-free mixing network. However, it is not possible to simply replace the IGM-based mixing network in the existing work with an IGM-free one, because this will cause each agent to fail to find the individual action corresponding to the optimal joint action during decentralized execution. Of course, this is why we propose the DAVE framework.

### D.2 Policy-Based MARL Methods on Single-State Matrix Games

DAVE is not a standard actor-critic framework in the strict sense. The ego policy in DAVE is updated in a supervised learning manner, which maximizes the probability of the joint action corresponding to the optimal joint action value in the sample. Thus DAVE can explicitly search for optimal joint actions and easily solve non-monotonic problems. FACMAC still has to rely on the monotonic mixing network to update the critic; otherwise, the performance will deteriorate.

Moreover, the existing policy-based method that does not use IGM for the critic cannot solve the non-monotonic task in our paper. We test two representative policy-based algorithms, MAPPO and MADDPG, in two didactic problems. We present the experimental results over 100 trials of 10000 episodes in Table 2 and Table 3, which demonstrate that although they do not explicitly introduce IGM, they still perform poorly.

|  | $k=0$ | | | $k=6$ | | | | $k=7.5$ | | | |
|---|---|---|---|---|---|---|---|---|---|---|---|
|  | -12 | 0 | **8** | -12 | 0 | 6 | **8** | -12 | 0 | 7.5 | **8** |
| MADDPG | 35% | 45% | 20% | 27% | 34% | 31% | 8% | 27% | 35% | 34% | 4% |
| MAPPO | 0% | 100% | 0% | 0% | 0% | 100% | 0% | 0% | 0% | 100% | 0% |

Table 2: Proportion of different convergence results in Matrix Game I.

|  | $k=0$ | | | $k=25$ | | | | $k=100$ | | | |
|---|---|---|---|---|---|---|---|---|---|---|---|
|  | 0 | 2 | **10** | -25 | 0 | 2 | **10** | -100 | 0 | 2 | **10** |
| MADDPG | 51% | 0% | 49% | 3% | 45% | 45% | 7% | 1% | 20% | 76% | 3% |
| MAPPO | 0% | 0% | 100% | 0% | 0% | 100% | 0% | 0% | 0% | 100% | 0% |

Table 3: Proportion of different convergence results in Matrix Game II.

We carry out MADDPG and MAPPO in a policy-based setting. The above two algorithms cannot perfectly solve the single-state matrix problems even when more samples can be accessed. As the difficulty of the problem increases, the proportion of MADDPG converging to the optimal solution gradually decreases. MAPPO converges steadily to a sub-optimal solution.

### D.3 Additional Explanations for Results in Multi-Step Matrix Game

In MAVEN's official code, the implementation of the multi-step matrix problem does not match the description in its original paper, which is reflected in the terminal state corresponding to the bad branch (see Figure 14). This does not change the non-monotonicity of the initial state, but reduces the expected return for agents to choose the bad branch. In other words, the above mistakes will make it **EASIER** for agents to learn the optimal policy. We rectified this error and retested all baselines for their real performance. Results in Figure 5 shows that MAVEN performed worse than in its original paper.

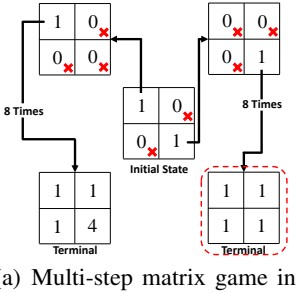

(a) Multi-step matrix game in MAVEN's paper.

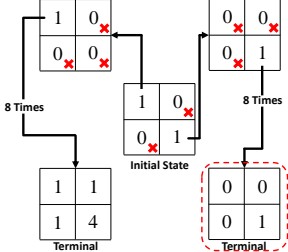

(b) Multi-step matrix game in MAVEN's code.

Figure 14: The difference between the multi-step matrix game environment in the original paper of MAVEN and in the actual code of MAVEN.

## D.4 Additional Introduction for SMAC Environment

The single-state matrix games and the multi-step matrix game are elaborated in the main body of the paper. Here we only elaborate on the SMAC environment. The action space of each agent is discrete. The version of StarCraft II used in this paper is the latest 4.10. For a fair comparison, all algorithms are run on the same version of StarCraft and SMAC. Table 4 shows the details of the scenarios used in this paper.

| Name | Ally Units | Enemy Units | Type |
|------|-----------|-------------|------|
| 2s_vs_1sc | 2 Stalkers | 1 Spine Crawler | Homogeneous Asymmetric |
| 3s5z | 3 Stalkers 5 Zealots | 3 Stalkers 5 Zealots | Heterogeneous Symmetric |
| 2c_vs_64zg | 2 Colossi | 64 Zerglings | Homogeneous Asymmetric |
| 5m_vs_6m | 5 Marines | 6 Marines | Homogeneous Asymmetric |
| 3s_vs_5z | 3 Stalkers | 5 Zealots | Homogeneous Asymmetric |
| 8m_vs_9m | 8 Marines | 9 Marines | Homogeneous Asymmetric |
| 10m_vs_11m | 10 Marines | 11 Marines | Homogeneous Asymmetric |
| MMM2 | 1 Medivac 2 Marauders 7 Marines | 1 Medivac 3 Marauder 8 Marines | Heterogeneous Asymmetric Macro tactics |
| 6h_vs_8z | 6 Hydralisks | 8 Zealots | Homogeneous Asymmetric |

Table 4: Maps in different scenarios.

## D.5 Additional Experiments on SMACv2

Although SMAC is a popular multi-agent benchmark in recent years, it suffers from a lack of stochasticity. To remedy this significant deficiency, a more challenging SMACv2 environment is proposed. In SMACv2, team composition is randomly generated, which means that unit types are not fixed. Meanwhile, the start positions of all agents are also random. Finally, the sight range and attack range of the units are changed to the values from SC2 instead of fixed values. These three major changes make the tasks in SMACv2 more challenging.

To assess the generalizability of our proposed algorithm, we evaluate the performance of DAVE in several representative scenarios of SMACv2. Some of the scenarios are symmetrical and others are asymmetrical. To ensure consistency with the original paper on SMACv2, we use the hyperparameter settings from PyMARL2[2] [15]. To demonstrate that the DAVE framework is applicable to any existing value decomposition method and can transform the original algorithm into an IGM-free version with guaranteed convergence, we compare the performance of two classic algorithms, QMIX and QPLEX, with their variants respectively. Specifically, IGM-Free variants are obtained by simply removing all absolute functions from the original methods. We conduct the experiments using StarCraft II version 4.10 and present the results in Figure 15 and Figure 16.

The experimental results demonstrate that DAVE-QMIX outperforms QMIX in most scenarios, particularly in the *protoss_5_vs_5* and *terran_5_vs_6* scenarios. Notably, both DAVE-QMIX and IGM-Free QMIX are IGM-free methods. However, while IGM-Free QMIX is unable to guarantee individual actions consistent with the optimal joint action, its performance is significantly worse than DAVE-QMIX and QMIX, and even fails in most scenarios. Similarly, DAVE-QPLEX outperforms both vanilla QPLEX and IGM-Free QPLEX, despite the weight of its transformation module no longer being restricted to non-negative values. In summary, DAVE variants perform significantly

---

[2]https://github.com/benellis3/pymarl2.

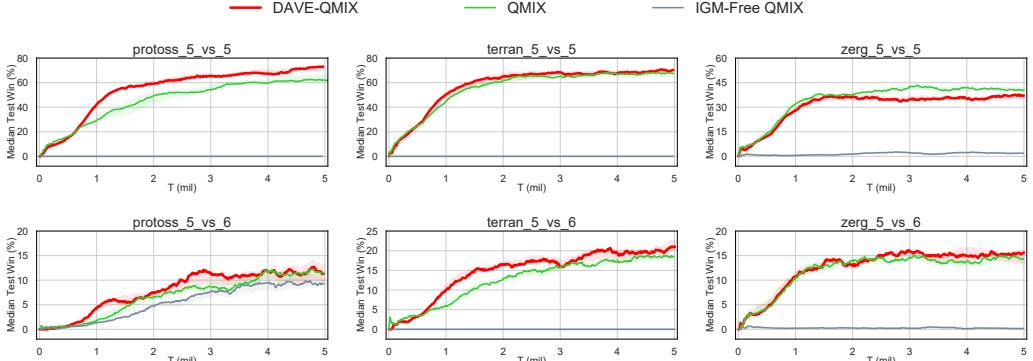

Figure 15: Comparisons of median win rate for variants of QMIX on SMACv2.

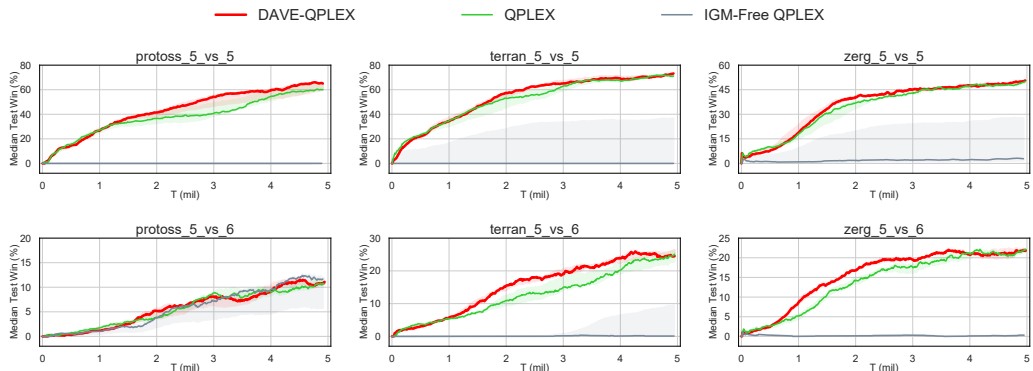

Figure 16: Comparisons of median win rate for variants of QPLEX on SMACv2.

better than original methods in scenarios that do not meet the IGM assumptions, and also exhibit competitive performance in other scenarios, regardless of the environment complexity.

In addition to evaluating different variants of the same algorithm, we compare the performance of different baselines designed to solve the IGM problem. Since these baselines are all based on QMIX, we show their learning curves and that of DAVE-QMIX in Figure 17. It is evident from the results that the complex mechanisms introduced by other algorithms cause them to perform worse than the vanilla QMIX and DAVE-QMIX. The experimental results on SMACv2 demonstrate the excellent generalization of DAVE to different algorithms and environments.

### D.6  Additional Experiments on Multi-Agent MuJoCo

In order to test the performance of the same algorithm following IGM and not following IGM in a complex environment, we also carry out vanilla QMIX, IGM-Free QMIX and DAVE on Multi-Agent MuJoCo (MA-MuJoCo). It is worth noting that the latter two do not require the IGM assumption at all, which means that they can fit richer function classes but it is more difficult to find the optimal joint action. Since most of the conventional value decomposition methods are only applicable to environments where the action space is discrete, we discretize the action space of the MA-MuJoCo environment reasonably. Each joint in the robot is considered as an agent. We use uniform quantization to discretize continuous actions and modify the action space of each agent to $\mathcal{U} = \left\{ \frac{2j}{K-1} - 1 \right\}_{j=0}^{K-1}$ instead of the original $\mathcal{U} = [-1, 1]$. $K$ is a hyperparameter and in this paper $K = 31$. In addition, MA-MuJoCo is a partially observable environment, and each agent (joint) can only observe the information of neighbor agents (joints). Figure 18 shows three classic scenarios. Numbers in brackets indicate the number of agents (joints) in each task.

It can be seen from Figure 19 that even without the IGM assumption, DAVE can still perform better than vanilla QMIX in most scenarios, especially in the *Hopper* and *Humanoid* scenarios. IGM-Free

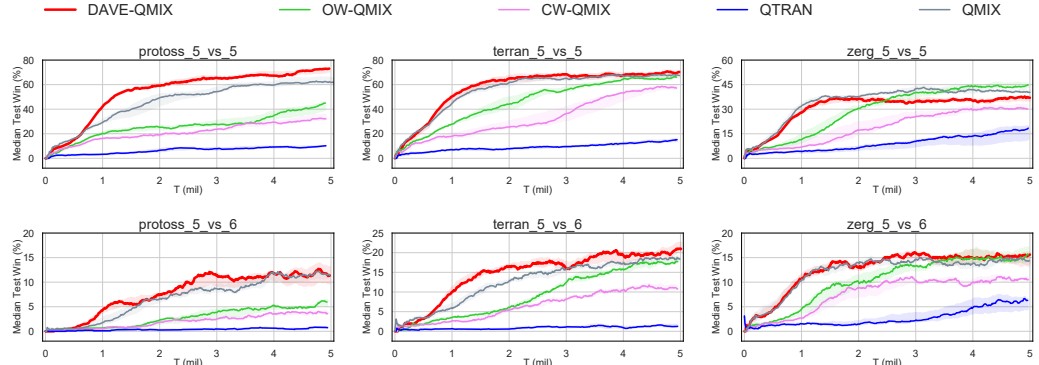

Figure 17: Median win rates on a range of SMACv2 mini-games. Note that only DAVE-QMIX is IGM-free.

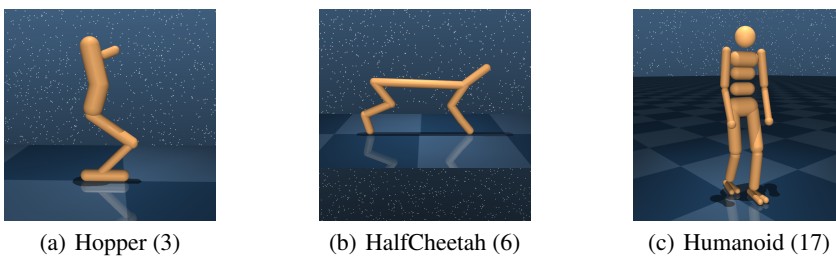

(a) Hopper (3)  (b) HalfCheetah (6)  (c) Humanoid (17)

Figure 18: Illustration of benchmark tasks in Multi-Agent MuJoCo.

QMIX fails in all tasks. In summary, in the case of the same loss function related to reinforcement learning, QMIX with the dual self-awareness framework performs better than vanilla QMIX in all cases. Even in scenarios with large action spaces like *Humanoid*, which contains $31^{17} \approx 2.26 \times 10^{25}$ joint actions, DAVE still outperforms QMIX. Like the results in the main text, the experimental results in MA-MuJoCo further prove that DAVE is not only suitable for didactic examples, but also capable of complex scenarios.

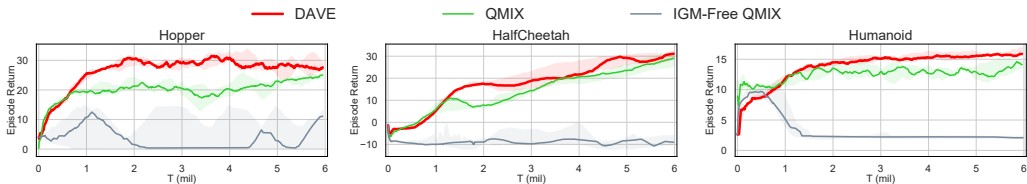

Figure 19: Median episode return on different MA-MuJoCo tasks. Note that only QMIX follows the IGM assumption.

### D.7 Additional Notes on Model Size

We give the relative size of neural network models for some value decomposition variants over QMIX in Table 5. The model size here refers to the number of parameters each algorithm needs to train in the *MMM2* scenario.

The number of parameters in the value decomposition framework mainly depends on the mixing network structure. The DAVE framework only has three more fully connected layers (an additional ego model) than QMIX, so DAVE has the smallest change compared to QMIX among the variants. Nevertheless, DAVE still performs better than other baselines related to IGM. The most important thing is that DAVE can quickly converge to the optimal solution in the matrix game, while QMIX cannot.

| Algo | QMIX | DAVE | CW-QMIX | OW-QMIX | MAVEN | QPLEX |
|---|---|---|---|---|---|---|
| **Relative Size** | 100% | 132% | 373% | 373% | 167% | 142% |

Table 5: The relative size of neural network models related to reinforcement learning for some value decomposition variants over QMIX.

# E   More Related Work

By improving the joint exploration of all agents, UneVEn [10] addresses the issue that stems from the monotonicity constraint in representing the joint-action value function. However, in some complex environments such as SMAC, its additional complex mechanism leads to significantly slower convergence. TESSERACT [22] views the Q-function as a tensor and utilizes low-rank tensor approximations to model agent interactions relevant to the task. DCG [2] factors the joint value function of all agents into payoffs between pairs of agents based on a coordination graph, but it requires additional domain knowledge and violates CTDE principles. QTRAN++ [31] is an improved version of QTRAN, however it still does not completely remove absolute functions. STEP [42] can solve non-monotonic problems but it focuses on how to find Stackelberg equilibrium via asynchronous action coordination. Besides, some value decomposition methods [33] from the distributional perspective replace the original IGM principle with the distribution-style IGM to deal with highly stochastic environments. The DAVE framework we proposed does not change the loss function of the underlying value decomposition algorithm, and only requires simple modifications to make it IGM-free. The DAVE-based value decomposition method can outperform the original algorithm in scenarios where the IGM assumption is not satisfied, and it can also have competitive performance in other scenarios.

There has been a lot of excellent work on exploration in the field of multi-agent reinforcement learning. UneVEn uses novel action-selection schemes during exploration to overcome relative overgeneralization. EITI/EDTI [38] measure the amount and value of one agent's influence on the other agents' exploration processes to improve exploration. Dimakopoulou and Roy [4] present several seed sampling schemes for efficient coordinated exploration in concurrent reinforcement learning. CMAE [18] uses target policies to maximize returns and exploration policies to collect unexplored data, which is similar to our work. However, CMAE focuses on solving the sparse-reward task, whereas anti-ego exploration in DAVE alleviates the suboptimal convergence caused by limited search. In addition, the anti-ego exploration mechanism also leverages the characteristics of the auto-encoder to further narrow the state-action space that needs to be explored.

