# OpenReview forum: "Dual Self-Awareness Value Decomposition Framework without Individual Global Max for Cooperative MARL"
_NeurIPS.cc/2023/Conference — NeurIPS 2023 poster_

### Official Review · Reviewer_BzQX · 2023-06-24

**Soundness:** 2 fair
**Presentation:** 1 poor
**Contribution:** 2 fair
**Rating:** 4
**Confidence:** 4

**Summary:**

The paper focuses on value decomposition without IGM constraints by proposing Dual self-Awareness Value dEcomposition (DAVE) framework. DAVE is inspired by dual self-awareness studied in psychology and uses an ego model, i.e., a policy for each agent for actual action selection and an alter ego model, i.e., a value function to address credit assignment without the IGM constraint. To avoid premature convergence to poor local optima, an anti-ego exploration method is proposed. DAVE is evaluated in several domains and compared with many state-of-the-art MARL approaches.

**Strengths:**

The paper proposes an interesting approach inspired by knowledge from psychology.

The evaluation is structured well by starting with small and tractable problems first and scaling up. Many state-of-the-art MARL algorithms are used as baseline for sufficient comparison.

**Weaknesses:**

The paper strongly focuses on the omission of IGM, which is motivated as a limitation of MARL. However, DAVE introduces additional complexity that at least doubles the computational effort compared to the baselines, e.g., two networks are used per agents and the exploration mechanism adds more parameters on top of that (note that despite doubled parameters, DAVE does neither perform twice as well nor learn twice as fast to fully compensate for the additional effort). Therefore, the fairness of comparison is questionable. Furthermore, I am not sure about the purpose of the individual Q-functions, since they are not used to maximize the joint Q function as usual. The ego model/policy is used instead thus individual Q-functions could be completely omitted as far as I understood the text.

The initial motivation of dual self-awareness in the introduction implies that ego and alter ego models have a symmetric relationship. However, the ego model represents a policy while the alter ego model represents a value function, which is sometimes called "alter ego model" and sometimes "alter ego value function model", which is confusing to read.

The maximization of the joint Q function through sampling, reminds me of FACMAC, where agents select their actions in a similar way (albeit for continuous action spaces though).

Despite the nice structure of the experimental presentation, the SMAC results are not particularly overwhelming because SMAC is a widely solved benchmark, where most baselines already achieve high win rates. Furthermore, the results are not in line with previous work, e.g.

- MAVEN reaches an average win rate of 40% at 5 million steps in 6h_vs_8z in the original paper, while it remains with 0% win rate in this paper with the same about of steps.
- QMIX reaches 100% win rate in 10m_vs_11m and 80% in 3s_vs_5z in the SMAC paper, while it is notably lower in this paper.

The paper heavily depends on the appendix indicating its lack of self-containment. The paper's content or the approach need more simplification to give a sense of completeness of the presentation.

Figure labels and numbers are only readable on a sufficient zoom scale which is problematic when printed.

**Questions:**

I do not fully understand the purpose of the alter ego models. Since the policies are trained to directly maximize the joint-Q function, why introducing a „detour“ to the mixing network through the individual Q-functions? Wouldn’t simply feeding the resulting joint action (e.g., as one hot vector) be sufficient to compute the TD-target for Eq. 4?

**Limitations:**

Some limitations like the additional compute through sampling actions are mentioned in the text.

Potential negative societal impact is not discussed at all.

---

> ### Author Rebuttal · Authors · 2023-08-02
>
> We thank the reviewer for the insightful and interesting comments! We are glad that you have read our paper carefully. We hope we can address your concerns below.
>
> **Q1**: Wouldn’t simply feeding the resulting joint action (e.g., as one-hot vector) be sufficient to compute the TD-target for Eq. 4?
>
> **A1**: We think this is a valuable question. We do not employ a monolithic critic whose input is the global state and the joint actions of all agents because our method can scale better to tasks with a larger number of agents and actions. For example, as shown in Figure 16(c), the *Humanoid* task involves 17 agents, each with 31 actions. Then the dimension of the sparse one-hot vector corresponding to the joint action is $17\times31=527$. However, if each agent has an alter ego model to output an individual Q-function as in DAVE, it can avoid generating the sparse high-dimensional one-hot vector mentioned above. In addition, we also tested the performance of DAVE when introducing sparse one-hot vectors corresponding to joint actions in complex environments. The performance did not change significantly but the algorithm lost scalability. So the value decomposition mechanism in DAVE is not a detour. We apologize for not explicitly mentioning this point in the main text and will incorporate it in the next version of our paper.
>
> **Q2**: Despite doubled parameters, DAVE does neither perform twice as well nor learn twice as fast to fully compensate for the additional effort.
>
> **A2**: We give the relative size of neural network models for some value decomposition variants over QMIX. The model size here refers to the number of parameters each algorithm needs to train in the *MMM2* scenario.
>
> | Algo          | QMIX | DAVE | CW-QMIX | OW-QMIX | MAVEN | QPLEX |
> | ------------- | :--: | :--: | :-----: | :-----: | :---: | :---: |
> | Relative Size | 100% | 132% |  373%   |  373%   | 167%  | 142%  |
>
> The number of parameters in the value decomposition framework mainly depends on the mixing network structure. The DAVE framework only has three more fully connected layers (an additional ego model) than QMIX, so DAVE has the smallest change compared to QMIX among the variants. Nevertheless, DAVE still performs better than other baselines related to IGM. The most important thing is that DAVE can quickly converge to the optimal solution in the matrix game, while QMIX cannot.
>
> **Q3**: The SMAC results are not particularly overwhelming because SMAC is a widely solved benchmark, where most baselines already achieve high win rates.
>
> **A3**: Our concern is that our IGM-free framework can achieve similar or better performance than IGM-based methods, especially in non-monotonic problems. In addition, we also tested the performance of DAVE and some baselines in other more challenging environments, including SMACv2 and MA-MuJoCo, which are shown in the appendix.
>
> **Q4**: The results are not in line with previous work.
>
> **A4**: We think there are two reasons. The first and most important point is that the version of StarCraft we use is 4.10 instead of 4.6.2 used in [1,2]. Performance is *not* always comparable between versions. Secondly, there are some small typos in the original paper of MAVEN. The X-axis in Figure 4(b) in [2] is strange, the "0m" scale is not at the origin and there are two "4m" scales. Also we point out some other typos about MAVEN in Appendix D.3. All experiments in this paper are carried out with 6 random seeds in order to avoid the effect of any outliers.
>
> **Q5**: The agents in FACMAC select their actions in a similar way as DAVE (albeit for continuous action spaces though).
>
> **A5**: The motivation of DAVE is completely different from that of FACMAC. FACMAC is IGM-based and focuses on continuous action space tasks, while DAVE is IGM-free and focuses on non-monotonic tasks. Although FACMAC proposes its IGM-free variant FACMAC-nonmonotonic, it performs worse than IGM-based methods in most of all scenarios, even in simple matrix games.
>
> We will try to simplify the paper's content while keeping the experiments sufficient and change the minimum text size in figures to 9pt. We really appreciate your comments and they really help us improve our paper. And we also appreciate it if you have any further comments or improve your score.
>
> **Reference**
>
> [1] Samvelyan, Mikayel et al. The StarCraft Multi-Agent Challenge. 2019.
>
> [2] Mahajan, Anuj et al. MAVEN: Multi-Agent Variational Exploration. 2019.

---

> > ### Comment · Reviewer_BzQX · 2023-08-19
> > **Answer**
> >
> > Thank you for your response. I have read it through and raised my score.

---

### Official Review · Reviewer_rpvy · 2023-07-02

**Soundness:** 2 fair
**Presentation:** 3 good
**Contribution:** 2 fair
**Rating:** 4
**Confidence:** 4

**Summary:**


This paper proposes a different approach for multi-agent RL algorithms that does not depend on individual global max (IGM), but instead builds a new framework, DAVE,  based on dual self-awareness. The algorithm is shown to perform favorably in several testing cases.


**Strengths:**

The paper is well organized and the sections’ structure serve their purpose in illustrating this new framework. The experimental results also corroborate the authors’ claims and show that the proposed framework/algorithm, DAVE, performs better than other MARL algorithms.


**Weaknesses:**

The main weakness in this paper is the lack of justification of differences between DAVE and the actor-critic models. In addition, since this paper proposes a new algorithm, it would be nice to have the pseudocode be in the main text. Overall, a better analysis of the sampling procedure is needed to justify the new DAVE framework.



**Questions:**

Question: from figure 2, it appears that the alter ego model calculates the q value function and is similar in concept to a critic network. The ego model also computes the policy which is the same as an actor network. How is the dual self-awareness model different from an actor-critic model? The description in lines 193-194 appears to limit the self-awareness model to be a special case of a general actor-critic model.

**Limitations:**


The proposed model “dual self-awareness” requires a specific sampling procedure to get the actor policy which renders this framework as a special case of actor-critic algorithms.

---

> ### Author Rebuttal · Authors · 2023-08-04
>
> We thank the reviewer for the sincere comments! We thank you for pointing out the many strengths of our paper. We hope we can address your concerns below.
>
> **Q1**: How is the dual self-awareness model different from an actor-critic model?
>
> **A1**: We apologize for misleading you into thinking that DAVE is a special case of actor-critic algorithms. We bolded "DISTINCT" in lines 193-194 to mean that DAVE is *completely* different from other actor-critic and policy-based algorithms. We then pointed out the reasons as follows:
>
> > The actions sampled from the ego policies remain independent and identically distributed (IID), so the ego policy is trained through *supervised learning* rather than *policy gradient* used in policy-based methods.
>
> All actors in the actor-critic framework are updated in the policy gradient manner, so the ego policies cannot be regarded as the actor model in the actor-critic framework. The ego policy model in DAVE is more like Best-of-N sampling, or rejection sampling [1] in the recently popular large language model field. We do not use policy gradient to update the ego policy because it is not effective for non-monotonic tasks, as shown in Appendix D.2. Details of the ego policy update can be found in Appendix B.
>
> Thank you so much for your helpful comments. And we also appreciate it if you have any further comments or improve your score.
>
> **Reference**
>
> [1] Touvron, Hugo et al. Llama 2: Open Foundation and Fine-Tuned Chat Models. 2023.

---

### Official Review · Reviewer_NJQv · 2023-07-06

**Soundness:** 3 good
**Presentation:** 3 good
**Contribution:** 3 good
**Rating:** 4
**Confidence:** 4

**Summary:**

This paper proposes a novel MARL algorithm, Dual self-Awareness Value dEcomposition (DAVE), which avoids the IGM constrain obeyed by most of the previous researches. The algorithm introduces three different policies including Ego Policy, Alter Ego Policy and Anti-ego Policy, where the Ego Policy tries to fit the optimal actions of the Alter Ego Policy and the Anti-ego Policy serves as an exploration strategy. Empirical results show that DAVE outperforms the baseline methods on several cooperative tasks.

**Strengths:**

1. The proposed DAVE algorithm avoids the IGM principle, which typically requires consistency between local and global optimal actions. This deviation enables DAVE to leverage the expressive capabilities of neural networks, particularly the mixing network component.
2. The Anti-Ego policy, a key component as the DAVE's exploration strategy, adopts the idea of count-based exploration techniques such as RND [1] and "Ensemble" in model-based RL. It quantifies the familiarity of $(s,a)$ pairs by evaluating the reconstruction loss of an auto-encoder.
3. Experimental results on Matrix Games show that DAVE does avoid local optima. Additionally, experiments conducted on SMAC, MA-MuJoCo indicate that DAVE-QMIX performs comparably to QPLEX and other baseline algorithms.


[1] Burda, Y., Edwards, H., Storkey, A., & Klimov, O. (2018). Exploration by random network distillation. arXiv preprint arXiv:1810.12894.

**Weaknesses:**

1. **Methodology**:

    *a*) The Ego Policy only approximates the global optimal actions $u^{*}$ by NLL loss, which means the equivalence between the Ego Policy and the global optimum is not strictly guaranteed while it is hold strictly by IGM principle.

    *b*) The basic idea of the exploration strategy is not quite novel. Simultaneously, while sampling from the Anti-Ego Policy guarantees the selection of low-Q actions, it does not guarantee that suboptimal-Q actions have been explored enough in the environment when the $M$ is not large enough. This is because reaching state $s$ is not guaranteed. As a result, the Q-values associated with these actions are prone to overestimation or underestimation, and the exploration policy tends to overlook this aspect of the action space.

2. **Experiments**: The results on SMAC and MA-MuJoCo are a bit confusing to me and i have listed my concerns in the *Qustions*.

**Questions:**

1. The optimization of the $Q_{\text{tot} }^\text{{alter}}$ network heavily relies on the action sample set $U^{\text{ego}}$ due to the computation of the target. However, unlike common MARL algorithms where the policy receives gradients through backpropagation, in DAVE, the Ego policy and $Q_{\text{tot}}^\text{{alter}}$ network are trained individually. Although *Proposition A.1* guarantees that objectively existing $u^*$ will be sampled when $M$ is sufficiently large, it does not ensure convergence to the optimal point in the optimization process.
Then, there are two different models need to optimize and they are highly related, how do you ensure the convergence of them?

2. There are concerns regarding whether Equation (8) truly returns "the most novel joint action." As mentioned in *1.b)* in the *Weakness* part, reaching state $s$ can be challenging. However, the action set $\hat{U}^{\text{ego}}$ is sampled from the Anti-Ego Policy. When $M$ is limited, the suboptimal-Q actions are hard to be sampled  both in the true interaction process (due to the hardness of reaching $s$) and in the Anti-Ego Policy sampling process (as their Q-values are not sufficiently low).

3.
    *a*) Most figures in the *Experiment Section* display the median performance. Can you replace them with the mean performance as it is more often used in other works?

    *b*) The results on SMAC of DAVE is based on QMIX as stated in the paper. To facilitate better comparison, it is suggested to use a more distinguishable color for the QMIX curve.

    *c*) If i've identified the QMIX curve correctly, DAVE exhibits significant superiority over QMIX in Figure 6. However, in the Appendix (Figure 13, 15, 17), DAVE shows only a slight advantage over QMIX, can you analyze the reason for this?

4. line 312: "The experimental results show that DAVE without anti-ego exploration still performs better than other baselines." Actually, I do not get this conclusion. In contrast, the green curve for "DAVE w/o Exploration" demonstrates a significant drop compared to the QMIX baseline. Does this mean that the exploration part plays the most important role in DAVE but not the relaxation of the IGM principle?

5. It is unclear why the curve for *M=100* does not outperform the others in Figure 7 (2s_vs_1sc, 5m_vs_6m). Further explanation is required to address this discrepancy.



**Limitations:**

Limitations: This work is mostly aimed to solve the cooperative Multi-agent tasks.

---

> ### Author Rebuttal · Authors · 2023-08-04
>
> We thank the reviewer for the detailed and constructive comments! We are glad that you have read our paper and the supplementary material carefully! We hope we can address your concerns below.
>
> **Q1**: The Ego policy and $Q_\text{tot}^\text{alter}$ network are trained individually. How to ensure the convergence of these two highly related models?
>
> **A1**: The training of these two models is complementary rather than independent. After evaluation by $Q_\text{tot}^\text{alter}$ network, the action $\boldsymbol{u}^\ast$ with high $Q_\text{tot}^\text{alter}$ will be used to train the ego policy. On the other hand, the new trajectory obtained by the ego policy interacting with the environment will help the $Q_\text{tot}^\text{alter}$ network to fit the real return. The above-mentioned update paradigm of DAVE is similar to Best-of-N sampling, also known as Rejection Sampling [1], which is an effective method in the recently popular large language model field.
>
> **Q2**: For the anti-ego exploration mechanism, it does not guarantee that suboptimal-Q actions have been explored enough in the environment when the $M$ is not large enough. Does Equation (8) truly return "the most novel joint action"?
>
> **A2**: Your analysis is very detailed and accurate. When $M$ is limited, it does affect the performance of the dual self-awareness framework and anti-ego exploration mechanism in DAVE. We illustrate this in both Figure 7 and Section 6. However, for most of the existing multi-agent benchmarks and real tasks, $M=100$ or even $10$ can make DAVE's performance close to or better than the existing IGM-based method. The resulting computational cost is entirely affordable.
>
> **Q3**: Can you replace the median performance in the Experiment Section with the mean one?
>
> **A3**: Thanks for your reminder. We will replace the aggregate function from median to mean.
>
> **Q4**: It is suggested to use a more distinguishable color for the QMIX curve in figures.
>
> **A4**: Thank you for your advice. We will correct it in the next version of our paper.
>
> **Q5**: DAVE exhibits significant superiority over QMIX in Figure 6 but shows only a slight advantage over QMIX in the appendix.
>
> **A5**: There are two main reasons. The first and most important reason is the illusion brought by the different ranges of the X-axis. In the appendix, we run all experiments for 5M or 6M steps. But in Figure 6, most tasks are run for 2M steps. This leads to the fact that DAVE's advantage over QMIX *appears negligible* in the appendix. This is a visual deception. For example, in the *terran_5_vs_6* task in Figure 15, DAVE can achieve 15% winning rate at 2M steps, while QMIX needs to achieve it at 3M.
>
> The second reason is that SMACv2 is more challenging than SMAC. Since the position, number, and category of agents are constantly changing in each episode, there may be situations where our agents are at a complete disadvantage. So there is no guarantee that the winning rate can theoretically reach 100%. Therefore, DAVE's performance can only be closer to a certain winning rate value than QMIX, but cannot exceed this value.
>
> **Q6**: I cannot draw such a conclusion that DAVE without anti-ego exploration still performs better than other baselines.
>
> **A6**: In Figure 5 Right, although the median of DAVE w/o Exploration has always been 10, its upper quantile reaches 12 first compared to other baselines. If we change the aggregation function from median to mean, the result will be more obvious. As shown in Figure 20 in the PDF we added, although the mean test return of DAVE w/o Exploration is lower than that of DAVE, it is still higher than other baselines. Furthermore, in the single-state matrix game, we implement various algorithms under uniform visitation to disregard the influence of exploration capabilities, as described in lines 269 to 270. Therefore,  in Figure 4, DAVE is equivalent to DAVE w/o Exploration.
>
> **Q7**: It is unclear why the curve for *M=100* does not outperform the others in Figure 7.
>
> **A7**: This discrepancy is due to the randomness of sampling. In *2s_vs_1c* scenario, the performance of DAVE with $M=100$ has been stable at 100% win rate while other performances are unstable. In addition, in *MMM2* map, the performance of DAVE with $M=100$ is also significantly better than others. In most scenarios, the difference in performance is not obvious, but this also shows that in common multi-agent tasks, $M=10$ or even $M=5$ is enough.
>
> **Q8**: The equivalence between the Ego Policy and the global optimum is not strictly guaranteed while it is held strictly by the IGM principle.
>
> **A8**: The IGM-based method cannot guarantee its strict convergence to the global optimum due to the limited family of functions. However, DAVE has a better chance of converging to the global optimal solution than the IGM-based method. This is why IGM-based methods tend to fall into local optimal solutions in matrix games, while DAVE can quickly converge to global optimal solutions.
>
> We really appreciate your comments and they really help us improve our paper. And we also appreciate it if you have any further comments or improve your score.
>
> **Reference**
>
> [1] Touvron, Hugo et al. Llama 2: Open Foundation and Fine-Tuned Chat Models. 2023.

---

### Official Review · Reviewer_uEFU · 2023-07-12

**Soundness:** 3 good
**Presentation:** 3 good
**Contribution:** 2 fair
**Rating:** 6
**Confidence:** 5

**Summary:**

This paper presents DAVE, an IGM-free value decomposition method, to enhance coordination ability in MARL. Drawing inspiration from the concept of dual self-awareness in psychology, DAVE consists of two components: an ego policy responsible for executing actions, and an alter ego value function involved in credit assignment and value estimation. By incorporating an explicit search procedure, DAVE eliminates the need for the IGM assumption and updates the actor using supervised signals instead of policy gradients. Additionally, the authors propose a novel anti-ego exploration mechanism to prevent the algorithm from being trapped in local optima. The method demonstrates impressive performance across a range of cooperative tasks, highlighting its effectiveness.

**Strengths:**

1. By employing NLL instead of policy gradients to update actors, DAVE effectively addresses the challenges associated with using unconstrained (without IGM assumption/monotonicity) factored critic.
2. The utilization of AE loss in Anti-Ego Exploration to assess the novelty of joint-action is intriguing, albeit bearing some similarities to RND [1].
3. The paper is well written and easy to follow. The integration of psychological concepts with algorithm design is a unique aspect that captured my interest. Furthermore, each component of the overall framework is thoroughly elucidated.
4. DAVE demonstrates impressive performance in both matrix games and SMAC, with clear discussions regarding its limitations.

**Weaknesses:**

1. While the authors claim that DAVE is the first method to completely eliminate the IGM assumption in value decomposition approaches, it's worth noting that FACMAC-nonmonotonic [2] also achieves full IGM-free. Although DAVE may outperform FACMAC-nonmonotonic, the contribution of DAVE might have been overestimated.
2. The discussion on related works is insufficient, and it would be beneficial to include references to value decomposition actor critic methods [2][5].
3. Minor 1: The idea of applying NLL loss bears similarities to previous works that combine Cross-Entropy Methods with Policy Improvement [3][4] (discussion can be included).
4. Minor 2: It appears that exploration has a significant impact on the performance of DAVE, making it a crucial component that cannot be disregarded, unlike simple epsilon-greedy exploration. It would be advisable to include comparisons between DAVE and other MARL exploration methods, such as EITI/EDTI[6] and CMAE[7], in the appendix.

**Questions:**

1. Considering that [2] states, "On SMAC, FACMAC-nonmonotonic performs similarly to FACMAC on easy maps, but exhibits significantly worse performance on harder maps," it would be worth discussing why the authors chose to compare DAVE with FACMAC-nonmonotonic instead of FACMAC (fairness may not be an issue).
2. There appears to be a notable discrepancy between the performance curves reported in Figure 6 of the current paper and Figure 7 in [2], despite both versions using SMAC 4.10. Could you explain the reasons behind this divergence?
3. While the elimination of the IGM assumption is touted as a significant contribution of DAVE, it would be valuable to explore the performance of DAVE with a monotonic mixing network (or others with IGM assumption). Does the unconstrained factored mixing network indeed enhance performance?

[1] Burda, Y., Edwards, H., Storkey, A., & Klimov, O. (2019, May). Exploration by random network distillation. In ICLR, 2019.

[2] Peng, B., Rashid, T., Schroeder de Witt, C., Kamienny, P. A., Torr, P., Böhmer, W., & Whiteson, S. (2021). Facmac: Factored multi-agent centralised policy gradients. *In NeurIPS, 2021*.

[3] Simmons-Edler, R., Eisner, B., Mitchell, E., Seung, S., & Lee, D. (2019). Q-learning for continuous actions with cross-entropy guided policies. *arXiv preprint arXiv:1903.10605*.

[4] Neumann, S., Lim, S., Joseph, A., Pan, Y., White, A., & White, M. (2018). Greedy Actor-Critic: A New Conditional Cross-Entropy Method for Policy Improvement. In ICLR, 2023.

[5] Wang, Y., Han, B., Wang, T., Dong, H., & Zhang, C. (2020). Off-policy multi-agent decomposed policy gradients. *arXiv preprint arXiv:2007.12322*.

[6] Ackermann, Johannes, et al. "Reducing overestimation bias in multi-agent domains using double centralized critics." *arXiv preprint arXiv:1910.01465* (2019).

[7] Wang, Tonghan et al. “Influence-Based Multi-Agent Exploration.” *ArXiv* abs/1910.05512 (2019)

**Limitations:**

As mentioned in Weakness.

---

> ### Author Rebuttal · Authors · 2023-08-05
>
> We thank the reviewer for the detailed and inspiring comments! We hope we can address your concerns below.
>
> **Q1**: Why did the authors choose to compare DAVE with FACMAC-nonmonotonic instead of FACMAC?
>
> **A1**: FACMAC is an IGM-based method. Because in the canonical implementation of FACMAC, the mixing network is a non-linear monotonic function, as in QMIX. And FACMAC focuses on continuous action space tasks rather than non-monotonic tasks. However, DAVE focuses on non-monotonic tasks, so the baselines we selected are methods related to relaxing the IGM assumption, except for the basic algorithm QMIX. The comparison of the performance of the IGM-based method and its DAVE variant on different tasks is most indispensable, so we compare DAVE-QMIX with QMIX, and DAVE-QPLEX with QPLEX in the appendix. The DAVE variant outperforms or equals its base algorithm in almost all scenarios. Thanks for your suggestion. We will add the comparison of DAVE-FACMAC and FACMAC in the next version of our paper.
>
> **Q2**: There appears to be a notable discrepancy between the performance curves reported in Figure 6 of the current paper and Figure 7 in the origin paper of FACMAC.
>
> **A2**: Figure 6 in our paper only contains 2 tasks that are also found in Figure 7 in the origin paper of FACMAC, namely *MMM2* and *2c_vs_64zg*. At the same time, only two algorithms are shared, namely QMIX and QPLEX. In our paper, the final winning rates of these two algorithms in these two scenarios are similar to those in Figure 7 in [1]. The difference in the training process may be due to the relatively large variance of the median.
>
> **Q3**: It would be valuable to explore the performance of DAVE with a monotonic mixing network.
>
> **A3**: Thank you very much for the reminder! This is indeed a valuable experiment. We added some experimental results, mainly the performance of DAVE with a monotonic mixing network on different domains. As shown in Figure 18 and 19 in the PDF we added in the global response, DAVE with a monotonic mixing network still cannot converge to the global optimal solution in matrix games. In addition, in the complex SMAC environment, the performance of DAVE with a monotonic mixing network is similar to or even better than that of QMIX, but worse than that of vanilla DAVE. This is also substantial proof that the unconstrained factored mixing network indeed enhances performance.
>
> **Q4**: Although DAVE may outperform FACMAC-nonmonotonic, the contribution of DAVE might have been overestimated.
>
> **A4**: Thank you for your correction. We will declare our contribution more rigorously. Before that, we declared that DAVE was the first method to completely eliminate the IGM assumption because we thought that FACMAC-nonmonotonic simply changed the mixing network of FACMAC from monotonic to non-monotonic. The resulting consequences caused by FACMAC-nonmonotonic have not been considered in depth. This also leads to the poor performance of FACMAC-nonmonotonic in most tasks.
>
> **Q5**: It would be beneficial to include some references and discussions.
>
> **A5**: Thank you for your suggestion. More discussion of related work can be found in Appendix E. And we will supplement the relevant literature and experiments you mentioned.
>
> We really appreciate your comments and they really help us improve our paper! And we also appreciate it if you have any further comments.
>
> **Reference**
>
> [1] Peng, B., Rashid, T., Schroeder de Witt, C., Kamienny, P. A., Torr, P., Böhmer, W., & Whiteson, S. (2021). Facmac: Factored multi-agent centralised policy gradients. *In NeurIPS, 2021*.

---

> > ### Comment · Reviewer_uEFU · 2023-08-15
> >
> > I thank the authors for their efforts and their response has addressed my concerns. I am currently maintaining my scoring.

---

### Official Review · Reviewer_Yoei · 2023-07-23

**Soundness:** 3 good
**Presentation:** 3 good
**Contribution:** 3 good
**Rating:** 7
**Confidence:** 4

**Summary:**

The paper proposes a novel value decomposition framework for MARL based on the notion of dual self-awareness in psychology. The framework, called DAVE, consists of two neural network models for each agent: the alter ego value function model and the ego policy model. The former participates in the evaluation of the group to solve the global credit assignment problem, and the latter helps the former to find the optimal individual action through an explicit search procedure. The paper claims that DAVE is the first value decomposition method that completely abandons the Individual Global Max (IGM) assumption, which requires consistency between local and global optimal actions. The paper also introduces an anti-ego exploration method to prevent the ego policy from getting stuck in a bad local optimum. The paper evaluates the performance of DAVE in both simple and complex environments, such as StarCraft II and Multi-Agent MuJoCo, and compares it to other popular baselines. The paper shows that DAVE can solve non-monotonic problems that challenge existing value decomposition methods, and can achieve competitive performance despite the absence of IGM. The paper argues that DAVE provides a new perspective on addressing the problems posed by the IGM assumption in value decomposition methods.

**Strengths:**

Improving the efficacy of CTDE and value decomposition-based approaches to cooperative MARL is an important topic. The paper's investigation of the role of the IGM principle in CTDE approaches is novel and should be of interest to the broader MARL and RL communities. The method is supported by empirical results in both small-scale, well-principled environments and larger-scale, complex environments; and is comprehensive in its comparison to baselines. The paper is well-written and clear.

**Weaknesses:**

- The related work in the main paper is a bit sparse given the depth of work in this area. Moving more of the discussion from Appendix E into the Related Work section would improve the contextualization of this work.
- Given the importance of sample size M, it would have been nice to see a greater exploration of the performance/computational cost trade-off. For example, how does M scale in more realistic environments? Are there environments where an unreasonably high value of M is needed? In the MMM2 example in Fig. 7 it looks like the learning signal is just beginning.
- There is limited discussion of limitations, future work, and societal impact.


**Questions:**

Can you provide more context into what is needed to tune exploration coefficients dynamically?
How well can this method generalize to other variants of the MARL problem (e.g. mixed incentive, social dilemmas)?
Can DAVE provably increase the solution concept reached by agents in social dilemmas / game theoretic settings?
How do you balance the trade-off between exploration and exploitation in the anti-ego exploration method?

**Limitations:**

The main limitation here is the computational cost of large values of M. This does not prevent strong performance in the environments studied in this work, but may be expensive in more complex and/or real-world environments. There are also other small limitations, e.g. tuning exploration coefficients.

---

> ### Author Rebuttal · Authors · 2023-08-06
>
> We thank the reviewer for the detailed and inspiring comments! We sincerely thank you for taking the time to read our paper and the supplementary material carefully! We hope we can address your concerns below.
>
> **Q1**: Can you provide more context into what is needed to tune exploration coefficients dynamically?
>
> **A1**: We propose an anti-ego exploration mechanism, which is able to select the action least likely to be selected by the ego policy in the current state $s$ (implemented by softmin) and the action least seen before (implemented by autoencoder). Through these two filterings, *relatively* unexplored actions can be effectively selected as long as the number of samples $M$ is large enough. In order to allow the agent to choose these unexplored actions in real interactions, we use NLL loss to increase the probability of selecting these actions, as shown in Equation 8. We use the exploration coefficient $\lambda$ to control the strength of exploration. We emphasize "relative" because almost all actions have been fully explored in the later stages of training, but there will always be actions that are *relatively* unexplored compared to others. Therefore, $\lambda$ must decrease over time and must eventually be 0, otherwise it will be unfavorable for exploitation in the later stage of training. Through some experiments, we found that the annealing period of $\lambda$ should account for 20%-40% of the entire training period. That is to say, anti-ego exploration is only performed during the first 20%-40% of the training period. A comparison of DAVE with different $\lambda$ initial values is shown in Figure 8.
>
> **Q2**: How well can this method generalize to other variants of the MARL problem (e.g. mixed incentive, social dilemmas)?
>
> **A2**: We focus on how to completely abandon the IGM principle, which is specific to value decomposition methods. The value decomposition method is only applicable to decentralized partially observable Markov decision process (Dec-POMDP) tasks, in which all agents share a reward function. Therefore DAVE cannot be directly applied to other variants of the MARL problem where agents have their own reward functions. In order to enable DAVE to quickly generalize to problems with mixed incentives or social dilemmas, there is a simple way to artificially design a global reward function related to all individual reward functions. In this way, DAVE is also able to converge to the optimal solution, just like the result of the matrix game in our paper.
>
> **Q3**: Can DAVE provably increase the solution concept reached by agents in social dilemmas / game theoretic settings?
>
> **A3**: Global reward game, such as Dec-POMDP task, can be formed as Markov game with a global reward. Its aim is learning a stationary joint policy so that no agent tends to unilaterally change its policy to maximize cumulative global rewards and a Markov equilibrium is reached. For example, for the two single-state matrix games in our paper, their corresponding Markov equilibrium are (A,A) and (A,C). As shown in Figure 4, DAVE can always quickly converge to the equilibrium solution while other IGM-based algorithms cannot. For other problems such as social dilemmas, we can try to convert the original problem into a Markov game with a global reward according to the method mentioned in **A2** and then solve it through DAVE.
>
> **Q4**: How do you balance the trade-off between exploration and exploitation in the anti-ego exploration method?
>
> **A4**: First of all, in order to allow the agent to exploit better, the exploration coefficient $\lambda$ is generally set to $\lambda\in[0,1)$ in Equation 8. The two terms in Equation 8 denote exploitation and exploration, respectively. Second, as mentioned in **A1**, the exploration in the later stage of training is less meaningful, so $\lambda$ should anneal over time and eventually become 0.
>
> **Q5**: It would have been nice to see a greater exploration of the performance/computational cost trade-off. For example, how does $M$ scale in more realistic environments? Are there environments where an unreasonably high value of $M$ is needed?
>
> **A5**: Thank you for your suggestion, we will add the experimental results in more realistic environments in the next version of the paper. According to our current version of the paper, we believe that $M=100$ is sufficient and affordable for several currently popular multi-agent benchmarks. It can be seen from Figure 7 that in most environments, when $M$ is $10$ or even $5$, the performance of DAVE does not drop or drops very little compared to $M=100$. Furthermore, the results for *Humanoid* task shown in Figure 16(c) also illustrate that $M=100$ is sufficient even for tasks with a larger number of agents and actions.
>
> **Q6**: Moving more of the discussion from Appendix E into the Related Work section would improve the contextualization of this work. And there is limited discussion of limitations, future work, and societal impact.
>
> **A6**: Thank you for your suggestion! We will correct it in the next version of the paper.
>
> We really appreciate your comments and they really help us improve our paper! And we also appreciate it if you have any further comments.

---

> > ### Comment · Reviewer_Yoei · 2023-08-15
> > **Thank you for the detailed response (no score change)!**
> >
> > Thank you to the authors for their detailed response -- they have clarified all of my questions and I am satisfied with their explanations.

---

### Author Rebuttal · Authors · 2023-08-08

Dear Reviewers,

Thank you for your time and effort in reviewing our manuscript. We are delighted to receive your comments and suggestions, which have been valuable in improving the quality of our research.

As part of the rebuttal process, we have submitted a PDF as an additional auxiliary material, containing supplementary information that we believe is important for your review. We would like to remind you to please take a moment to review the attached file for further details. The PDF contains the following content:

1. The learning curves of vanilla DAVE and DAVE with a monotonic mixing network on the matrix games and SMAC.

2. Performance over time on the multi-step matrix game. As suggested by Reviewer NJQv, we change the aggregation function from *median* to *mean* in Figure 5.

We have also addressed each comment and provided responses inline with the reviewers' comments. We believe that these responses fully address the concerns raised by each reviewer and hope that you find them satisfactory.

We look forward to hearing your feedback and appreciate the time you have taken to review our manuscript. Thank you for your consideration.

---

### Decision · Program_Chairs · 2023-09-21

**Decision:**

Accept (poster)

**Comment:**

The paper introduces DAVE, a novel value decomposition framework for multi-agent reinforcement learning inspired by the dual self-awareness concept in psychology. DAVE does not use the Individual Global Max (IGM) assumption, offering a fresh approach to addressing issues linked to the IGM assumption in value decomposition.

The authors have addressed the reviewers' main concerns. The mitigation of the IGM assumption/monotonicity is an important challenge for the MARL community. While the lower scoring reviewers did not change their scores, they did not respond to the authors' as well as further attempts to understand if the authors have addressed their concerns. I find most of the reviewers' concerns to be simple to address. I therefore recommend acceptance of the paper, given that they will address the reviewers' comments in the final version of their paper.